# Presynaptic developmental plasticity allows robust sparse wiring of the *Drosophila* mushroom body

**Najia A Elkahlah[1†], Jackson A Rogow[2†], Maria Ahmed[1], E Josephine Clowney[1]***

[1]Department of Molecular, Cellular and Developmental Biology, The University of Michigan, Ann Arbor, United States; [2]Laboratory of Neurophysiology and Behavior, The Rockefeller University, New York, United States

**Abstract** In order to represent complex stimuli, principle neurons of associative learning regions receive combinatorial sensory inputs. Density of combinatorial innervation is theorized to determine the number of distinct stimuli that can be represented and distinguished from one another, with sparse innervation thought to optimize the complexity of representations in networks of limited size. How the convergence of combinatorial inputs to principle neurons of associative brain regions is established during development is unknown. Here, we explore the developmental patterning of sparse olfactory inputs to Kenyon cells of the *Drosophila melanogaster* mushroom body. By manipulating the ratio between pre- and post-synaptic cells, we find that postsynaptic Kenyon cells set convergence ratio: Kenyon cells produce fixed distributions of dendritic claws while presynaptic processes are plastic. Moreover, we show that sparse odor responses are preserved in mushroom bodies with reduced cellular repertoires, suggesting that developmental specification of convergence ratio allows functional robustness.

**\*For correspondence:**
jclowney@umich.edu

[†]These authors contributed equally to this work

**Competing interests:** The authors declare that no competing interests exist.

## Introduction

The environmental stimuli animals encounter on a day-to-day basis are extraordinarily numerous. Olfactory systems have evolved to cope with this diversity by maximizing the chemicals that can be detected, through the amplification of chemosensory receptor gene families, and through combinatorial coding, which expands representation capacity from the number of receptors in the genome to the number of combinations among them. The arthropod mushroom body is a cerebellum-like associative learning structure with a well-understood role in representing sensory stimuli and associating sensory and contextual cues (*Farris, 2011*; *Hige, 2018*; *Kennedy, 2015*). While mushroom bodies of different insect species process information from a variety of sensory modalities, 90% of Kenyon cell inputs in *Drosophila melanogaster* are olfactory (*Zheng et al., 2018*). The mushroom body of each hemisphere has ~2000 Kenyon cells (KCs), which are two synapses from the sensory periphery. Each olfactory receptor neuron in the antennae of adult flies expresses one or two of 75 olfactory receptor genes encoded in the genome. The axons of neurons expressing the same receptor converge on one of 54 glomeruli in the antennal lobe. ~150 uniglomerular projection neurons (PNs) have dendrites in one of the 54 glomeruli and carry signals about distinct receptor channels to two regions of the protocerebrum, the lateral horn and the mushroom body calyx (*Figure 1A*). PN inputs to the lateral horn are thought to underlie innate behaviors, while inputs to the mushroom body allow flexible learned association of odor stimuli with behavioral context (*Chin et al., 2018*; *de Belle and Heisenberg, 1994*; *Fişek and Wilson, 2014*; *Jefferis et al., 2007*; *Ruta et al., 2010*).

In the mushroom body calyx, the presynaptic sites of individual olfactory PNs cluster into multi-synaptic boutons, with PNs of different types (innervating different glomeruli) producing consistent, characteristic bouton numbers (*Caron et al., 2013*; *Zheng et al., 2018*). Each PN makes 1–20

**eLife digest** Despite having a limited number of senses, animals can perceive a huge range of sensations. One possible explanation is that the brain combines several stimuli to make each specific sensation.

The olfactory learning system in the fruit fly *Drosophila melanogaster* is in a part of the brain called the mushroom body. It allows fruit flies to associate a specific smell with a reward (e.g. food) or a punishment (e.g. poison) and behave accordingly. Two groups of neurons process stimuli from sensory receptors in the mushroom body: olfactory projection neurons carry information from the receptors and pass it on to neurons called Kenyon cells. The system relies on Kenyon cells receiving the combined input of multiple olfactory projection neurons, and therefore information from multiple receptors. The number of inputs each Kenyon cell receives is thought to determine the number of sensations that can be told apart, and thus, the number of signals that can be used for learning.

While many mechanisms dictating the complexity of a neuron's shape have been described, the logic behind how two populations of neurons become connected to combine several inputs into a single sensation has not been addressed. A better understanding of how these connections are established during development can help explain how the brain processes information, and the *D. melanogaster* mushroom body is a good system to address these questions.

Elkahlah, Rogow et al. manipulated the number of olfactory projection neurons and Kenyon cells in the mushroom body of fruit flies during development. They found that despite there being a varying number of cells, the number of connections into a post-synaptic cell remained the same. This indicates that the logic behind the combinations of inputs required for a sensation depends on the Kenyon cell, while olfactory projection neurons can adapt during their development to suit these input demands. Thus, if there are fewer Kenyon cells, the olfactory projection neurons will each provide connections to fewer cells to compensate, and if there are fewer olfactory projection neurons, each of them will input into more Kenyon cells. To show that the developing mushroom body could indeed adapt to different numbers of olfactory projection neurons and Kenyon cells, the modified flies were tested for olfactory perception: their responses to odor were largely normal.

These results underline the robustness of neuronal circuits. During development, the mushroom body can compensate for missing or extra neurons by modifying the numbers of connections between two groups of neurons, thus allowing the olfactory system to work normally. This robustness may also predispose the system to evolutionary change, since it allows the system to continue working as it changes. These findings are relevant to any area of the brain where neurons rely on combined input from many sources.

---

boutons, and each bouton is wrapped by claws of ~10 KCs, such that each PN sends output to between 10 and 200 of the 2000 KCs (*Leiss et al., 2009*). KCs in turn have 3–10 (average of five) claws, which innervate boutons of various PNs (*Caron et al., 2013*; *Gruntman and Turner, 2013*; *Zheng et al., 2018*). Each KC therefore receives innervation from only a minority of the 54 incoming sensory channels, and different individual KCs receive different and relatively unstructured combinations of inputs (*Caron et al., 2013*; *Eichler et al., 2017*; *Honegger et al., 2011*; *Murthy et al., 2008*; *Zheng et al., 2018*). The sets of inputs to individual cells vary across hemispheres and likely across individuals (*Caron et al., 2013*; *Eichler et al., 2017*; *Honegger et al., 2011*; *Murthy et al., 2008*). Associative learning mechanisms operate at KC output synapses, in the mushroom body axonal lobes, to re-weight KC outputs depending on experience and shift animal behavior (*Cohn et al., 2015*; *Handler et al., 2019*; *Hige et al., 2015*; *Owald and Waddell, 2015*).

The mushroom body is a simplified and experimentally tractable example of an expansion layer, in which a set of sensory inputs is mapped combinatorially onto a much larger set of postsynaptic cells, increasing the dimensionality of sensory representations. Like the diversification of antibodies by V(D)J recombination, the diversification of sensory input combinations across KCs is thought to allow them to represent arbitrary odors, regardless of evolutionary experience. Neurons of many other expansion layers receive similarly few, or sparse, sensory inputs. These include the cerebellum proper, the electric organ of mormyrid fish, the dorsal cochlear nucleus, and the hippocampus

(*Bell et al., 2008*; *Keene and Waddell, 2007*; *Mugnaini et al., 1980*). Cerebellar granule cells have an average of four large, claw-shaped dendrites that are innervated by clustered mossy fiber presynaptic sites in mossy fiber rosettes. The similar convergence ratios in the cerebellum and mushroom body (4 or 5 sensory inputs, respectively, per expansion layer cell) are thought to maximize dimensionality of sensory representations by optimizing the tradeoff between stimulus representation, which is maximized when expansion layer neurons receive large combinations of inputs, and stimulus separation, which is maximized when expansion layer neurons receive few inputs (*Albus, 1971*; *Cayco-Gajic et al., 2017*; *Litwin-Kumar et al., 2017*; *Marr, 1969*). The number of sensory inputs received by expansion layer neurons is thus a crucial parameter in sensory coding. How the density of inputs to expansion layer neurons is developmentally programmed is not understood in any system.

Innervation complexity more generally has been studied in the peripheral nervous system and in the developing mammalian cortex. In peripheral sensory neurons, most prominently those of the *Drosophila* larval body wall, cell-autonomous mechanisms profoundly influence dendritic complexity (*Corty et al., 2016*; *Jan and Jan, 2010*; *Ziegler et al., 2017*). However, sensory neurons do not need to coordinate their innervation with presynaptic partners. In the vertebrate peripheral nervous system, including the rabbit ciliary ganglion and vertebrate neuromuscular junction, postsynaptic neurons or muscles are thought to dictate wiring complexity (*Gibbins et al., 2003*; *Hume and Purves, 1983*; *Hume and Purves, 1981*; *Purves and Hume, 1981*; *Turney and Lichtman, 2012*). In contrast, in the developing cortex, extracellular signals including BDNF play a strong role in influencing dendritic complexity, suggesting that presynaptic cells and glia also influence connectivity density (*Kohara et al., 2003*; *McAllister et al., 1997*; *McAllister et al., 1995*; *Purves, 1986*). Therefore, while mechanisms acting in both pre- and post-synaptic cells *can* influence innervation complexity, there is a need to directly compare how pre- and post-synaptic cells influence one another.

We sought to ask how convergence ratio is set in the mushroom body calyx. By bidirectionally varying the populations of pre- and post-synaptic cells, we were able to make many different mushroom body aberrations. Across these conditions, we found a consistent pattern of compensations: the number of claws per KC remained largely constant, while the number of presynaptic boutons per olfactory PN varied bidirectionally and in response to changes of both the PN and KC populations. We therefore conclude that in this circuit, connectivity density is set by aspects of KC specification and is accomplished by flexible innervation of the calyx by PNs.

## Results

In order to ask whether presynaptic olfactory PNs or postsynaptic KCs (or both) dictate sparse wiring density in the mushroom body calyx, we used a variety of methods to manipulate the ratio between pre- and post-synaptic cells during embryonic or larval stages, well before adult connections form in the pupa. Through a combination of genetic and pharmacological interventions, we were able to vary both cell populations up and down. In these four conditions, we then examined gross calyx anatomy in adult animals and, when possible, the processes of individual PNs and KCs. These experiments allowed us to ask which population of neurons would adjust its production of processes in response to the perturbation. As we will describe, in all four cases, PNs adjusted their repertoire of presynaptic sites while KC claw number remained similar to that of unmanipulated animals.

### Projection neurons reduce bouton number when Kenyon cell number is reduced

First, we sought to reduce the KC population, to ask whether remaining cells would increase their claw number to fully innervate incoming PN boutons, or whether PNs would scale down their boutons to the KC repertoire. To do this, we took advantage of existing pharmacological techniques for KC neuroblast ablation (*de Belle and Heisenberg, 1994*; *Sweeney et al., 2012*). In the fly, four KC neuroblasts in each hemisphere produce ~500 KCs each (*Ito et al., 1997*). While most neuroblasts pause their divisions for the first 8 hr after larval hatching (ALH), the KC neuroblasts continue to divide; if larvae are fed the mitotic poison hydroxyurea (HU) during this time, KC neuroblasts can be specifically ablated (*de Belle and Heisenberg, 1994*). In these animals, lacking all KCs that receive olfactory inputs, olfactory learning is abrogated while innate olfactory behaviors are spared (*de Belle and Heisenberg, 1994*). The uniglomerular, excitatory olfactory PNs that innervate the MB calyx are

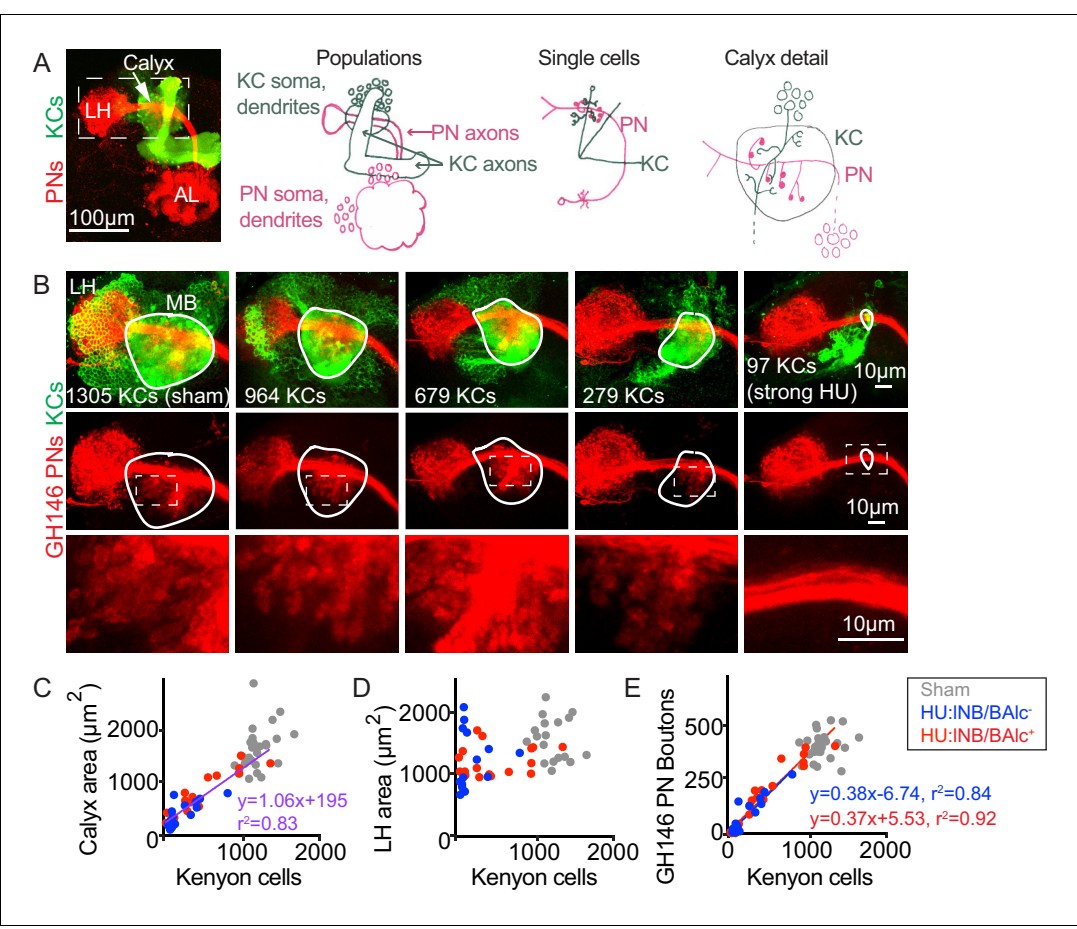

**Figure 1.** Reduction of Kenyon cells leads to reduced projection neuron innervation of the mushroom body calyx. (A) Image (left) and models (right) of olfactory PNs and KCs in the adult fly brain. PNs (red) receive input from olfactory sensory neurons in the antennal lobe (AL) and project to the mushroom body calyx and the lateral horn (LH). In the calyx, PNs synapse with KCs (green) in microglomeruli. Dashed box at left indicates region shown in B. (B) Maximum intensity projections of confocal stacks of the brains of adult flies subjected to HU treatment or sham treatment just after larval hatching. PNs labeled red using GH146-Gal4, KCs green using MB247-LexA. White outlines indicate the extent of the MB calyx in each sample; dashed boxes are regions shown magnified below. (C, D) Relationship between number of KCs and calyx (C) or lateral horn (D) maximum cross-sectional area in HU treated (blue, red dots) and sham treated (gray dots) MBs. Treated samples are separated by presence (red) or absence (blue) of lNB/BAlc progeny. Here and throughout, each dot is one hemisphere. Purple trend line in (C) combines all HU-treated samples. (E) Relationship between KC number and GH146[+] bouton number in brains of HU- and sham-treated animals.

The online version of this article includes the following video and figure supplement(s) for figure 1:

**Figure supplement 1.** Relationship between KC number, calyx volume, and calyx area.

**Figure supplement 2.** Method of bouton counting.

**Figure 1—video 1.** Volume renderings of mushroom bodies shown in *Figure 1B*.

https://elifesciences.org/articles/52278#fig1video1

produced by two neuroblasts, called lNB/BAlc and adNB/BAmv3 . Besides the KC neuroblasts, lNB/BAlc is the only other neuroblast in the fly central nervous system dividing from 0 to 8 hr ALH and is thus also susceptible to HU ablation (*Das et al., 2013*; *Ito and Hotta, 1992*; *Lovick et al., 2016*; *Lovick and Hartenstein, 2015*; *Stocker et al., 1997*; *Sweeney et al., 2012*).

While ablation of all eight KC neuroblasts (4/hemisphere) was used to demonstrate the role of KCs in olfactory learning (*de Belle and Heisenberg, 1994*), we sought to perform more subtle manipulations of the mushroom body cell populations. We generated animals in which PNs and KCs were fluorescently labeled to facilitate scoring, and reduced the concentration and duration of HU

application, producing mushroom bodies with a variety of Kenyon cell complements, likely due to sporadic loss of different neuroblast complements in each hemisphere (*Figure 1B*).

Together, these neuroblast losses could potentially produce 1–2 PN neuroblasts and mushroom bodies derived from 0 to 4 KC neuroblasts. Using our reporters, we scored the presence or absence of PNs ventrolateral to the antennal lobe, derived from lNB/BAlc, and counted the number of KCs (*Figure 1C*). In practice, some of the neuroblast states did not occur and some were overrepresented (*Figure 2—figure supplement 1*). Hemispheres from the same brain did not generally show the same neuroblast state as one another, but were often similar in severity of neuroblast losses (*Figure 2—figure supplement 1*).

KCs of different types are produced in a characteristic sequence in development: γ KCs are born in the embryo and early larva, α'β' KCs are born in late larvae, and αβ KCs are born in pupae. We identified samples in which all 4 KC neuroblasts had been ablated by looking for samples lacking mushroom body vertical lobes, which are composed of axons of α'β' and αβ KCs and thus do not form if all KC neuroblasts are killed by HU just after hatching. We confirmed previous findings that in animals with all four KC neuroblasts ablated, the PNs fail to target the calyx and project only to the lateral horn (*Stocker et al., 1997*). In these animals, ~100 KCs remained (*Figure 1B*). These represent the KC population that is born prior to larval hatching, and which is thus unaffected by KC neuroblast ablation after hatching. Recently, these earliest-born cells, called γd KCs, have been shown to receive visual and not olfactory input in adult flies (*Vogt et al., 2016*; *Yagi et al., 2016*). These observations suggest that γd KCs are molecularly distinct from later-born cells during pupal calyx wiring, such that even when all the olfactory KCs are absent, olfactory PNs cannot innervate visual KCs. Remarkably, these embryonic-born γd KCs receive olfactory inputs in the early larva and are sufficient for olfactory learning at that time (*Eichler et al., 2017*; *Pauls et al., 2010*); they must therefore switch their partner preferences across developmental stages.

We then examined calyx anatomy in animals with intermediate populations of KCs, likely derived from 1 to 3 remaining neuroblasts. We found a progressive decrease in the size of the mushroom body calyx in these animals as measured by the maximum cross-sectional area of the calyx (*Figure 1B,C*) or by calyx volume (*Figure 1—figure supplement 1*, *Figure 1—video 1*). To ask if this corresponded to a decline in bouton number, we counted the number of PN boutons in each hemisphere (*Figure 1E*, *Figure 1—figure supplement 2*). This revealed a linear relationship between KC number and PN bouton number, suggesting that PNs reduce their population of boutons to match the KC population. The lateral horn appeared normal in these animals and its size did not correlate with KC number, suggesting that PN projections to these two brain areas develop independent of one another (*Figure 1B,D*). The presence or absence of ventrolateral PNs did not obviously predict bouton number, suggesting individual PNs may be able both to tailor their bouton production to the number of KCs and to adjust to the number of other PNs. However, we were not able to obtain enough samples to draw firm conclusions about whether PN cell number affected PN bouton number in calyces derived from matched KC neuroblast complements. To test this, we developed alternative methods to ablate PNs, described below.

To score KC clones directly, we developed a strategy to label only the latest-born, 'αβ core' KCs. The somata and principle neurites of these late-born cells remain clustered in the adult, allowing us to count KC clones by counting groups of labeled soma or axon tracts as they leave the calyx (*Figure 2A,B*). Again, we found that calyx size tracked KC clone number, and that the presence or absence of the PN progeny of lNB/BAlc did not predict calyx size (*Figure 2C*). KC loss could reduce PN bouton number because PNs die or fail to be born when lacking KC contacts or trophic support. We thus counted the number of anterodorsal PNs (derived from adNB/BAmv3) and ventrolateral PNs (derived from lNB/BAlc) in this cohort. The number of anterodorsal PNs was not reduced in animals with some or all KC neuroblasts ablated, and the number of ventrolateral PNs was binary rather than graded, suggesting that vlPN number is determined by whether lNB/BAlc succumbed to HU, rather than by the KC complement (*Figure 2D,E*). These results suggest that PN neurogenesis and survival do not require KC signals. Like γd KCs, a few PNs derived from lNB/BAlc remained in adults; these likely represent PNs born embryonically from lNB/BAlc.

Reduction in the number of PN boutons in the calyx could occur because individual PNs quantitatively reduce their bouton production. To ask whether individual PNs alter their production of presynaptic boutons as the KC repertoire changes, we identified two methods to label single PNs. We focused on anterodorsal PNs, as these are not susceptible to ablation of lNB/BAlc. First, we used

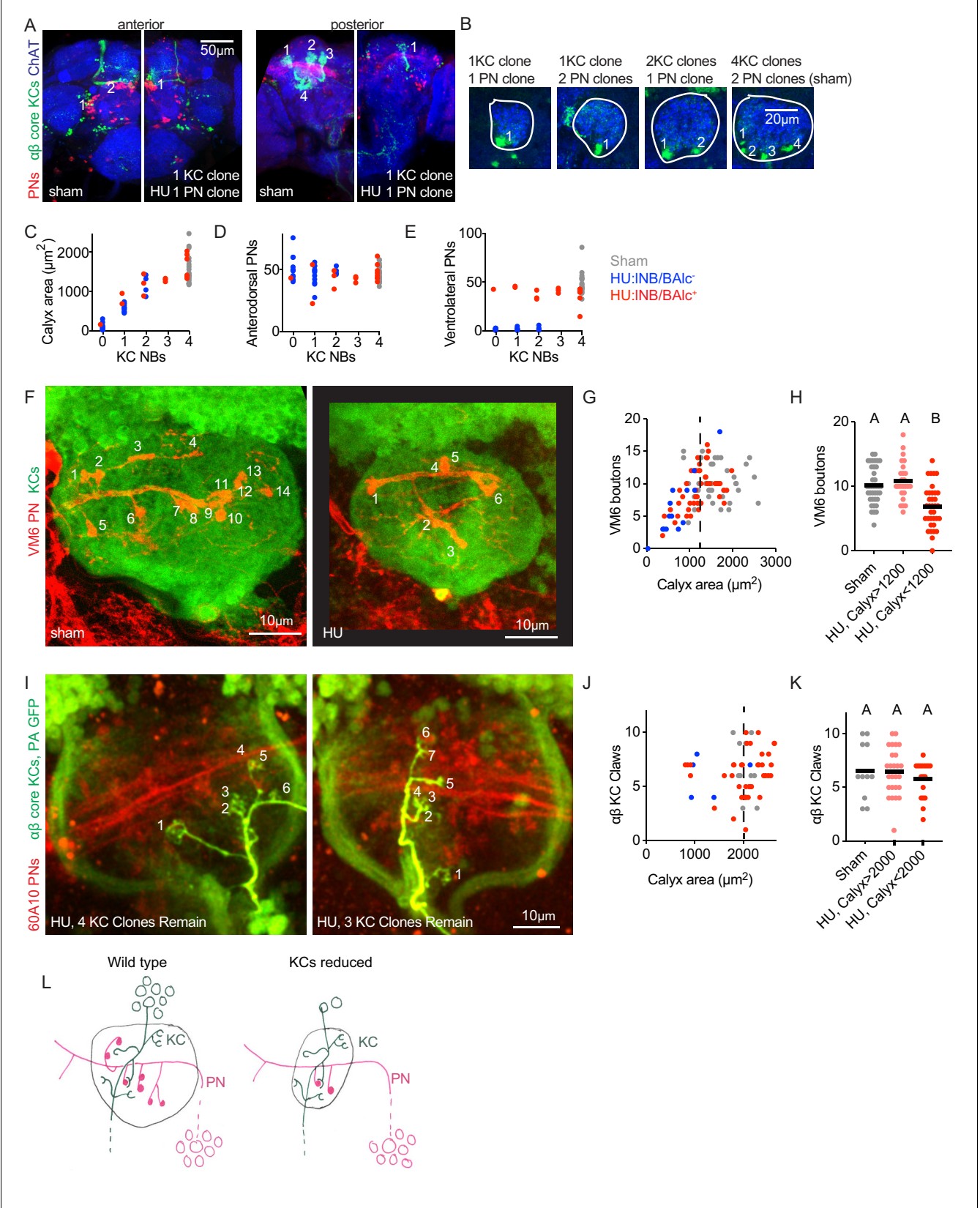

**Figure 2.** Individual projection neurons scale their bouton repertoire to the Kenyon cell population. (**A**) Maximum intensity projections of confocal stacks of the anterior (left) and posterior (right) of HU and sham-treated animals. GH146 labels PNs (red). 58F02 labels late-born αβ 'core' KCs (green). *Figure 2 continued on next page*

Figure 2 continued

Cholinergic neurons labeled with ChAT immunostaining (blue). Numbers indicate PN and KC clones. Anterior and posterior images are of the same brains. (B) Confocal slices of the mushroom body calyx in sham-treated (right) and various HU treated (left) hemispheres. ChAT signal highlights bouton structures, while 58F02 signal allows scoring of the number of KC clones. Numbered neurite bundles innervate the pedunculus when traced through the stack, while green signals not numbered derive from non-KC cell types labeled by 58F02. (C–E) Relationship between KC clone number and maximum calyx cross-sectional area (C), and number of anterodorsal (D) and ventrolateral (E) PNs. (F–H) Example images (F) and quantification (G, H) of MB calyx boutons on the VM6 PN, labeled by 71D09 (red). KCs labeled by MB247 (green). Numbers in (F) indicate counted boutons, and these two calyces are displayed at the same scale. Dashed vertical lines in (G) indicate cutoff between HU-treated calyces similar to controls and those affected by the manipulation. Significance, one-way ANOVA with correction for multiple comparisons. Here and throughout, letters above scatterplots indicate statistically distinguishable versus undistinguishable groups by ANOVA. Black horizontal bars represent medians. (I–K) Example images (I) and quantification (J, K) of GFP photoactivation of individual KCs in HU treated or sham brains. PNs labeled by 60A10 and αβ 'core' KCs by 58F02. Examples in (I) show two hemispheres of the same HU treated brain. Note that calyx area is larger in brains imaged ex vivo by two photon (as in I-K) versus fixed, stained, and imaged by confocal (as in F-H). Quantification in (J, K) includes αβ KCs labeled as shown in (I) and those labeled as shown in *Figure 2—figure supplement 3*; quantification of γ KCs labeled by both methods is shown in *Figure 2—figure supplement 3*. lNB/BAlc status was scored by examining PN somata near the antennal lobe, as in (A). (L) Model of the effect of KC loss on calyx development. Remaining KCs exhibit wild-type numbers of claws, while PNs reduce their bouton repertoire, reducing the size of the calyx.

The online version of this article includes the following figure supplement(s) for figure 2:

**Figure supplement 1.** Distribution of HU effects.

**Figure supplement 2.** Images and quantification of individual 42D01-labeled PNs highlighted by GFP photoactivation.

**Figure supplement 3.** Additional examples and quantification of KCs labeled by GFP photoactivation.

71D09-Gal4 to label the anterodorsal VM6 PN (*Ward et al., 2015*) and then subjected labeled animals to HU ablation. Because we found a linear relationship between KC number and calyx area (*Figures 1C, 2C*), we used calyx area as a proxy for KC number. While VM6 usually produces ~10 boutons, we found only ~5 boutons in animals with reduced calyx size, and 0–1 bouton in animals with severe calyx reductions, likely lacking all olfactory KCs (*Figure 2F–H*). These results suggest that individual PNs do reduce their bouton production as the KC complement is reduced, and that this reduction is graded rather than binary (i.e. in animals with some but not all KC neuroblasts ablated, bouton number is in between 0 and the wild type number). To characterize additional PNs, we expressed photoactivatable GFP under control of R42D01-Gal4, which labels ~10 anterodorsal PNs innervating the VM3 and VM4 glomeruli. We photoconverted individual somata using two photon microscopy (*Figure 2—figure supplement 2*). Though bouton counts were somewhat more variable than for VM6 as this strategy labels different types of PNs with different characteristic bouton numbers, we again found a correlation between KC loss and bouton reduction (*Figure 2—figure supplement 2*).

The graded reductions in bouton number we observed for different PN types would preserve the relative representation of these odor channels in the calyx. We also note that the range of bouton numbers we observed are consistent with adjustments to compensate for the presence or absence of ventrolateral PNs. For example, most VM6 neurons innervating a calyx derived from 1 KC neuroblast produced between 3 and 7 boutons, as compared to the median of 10 in unperturbed animals. If KCs are reduced by ¾ without loss of lNB/BAlc, perfect compensation by PNs would reduce boutons from 10 to 2.5. If lNB/BAlc was also lost, perfect compensation would reduce boutons from 10 to 5. These predicted values are similar to the bouton range we observed experimentally.

Finally, we sought to ask if KC claw number changes as KC number changes. We used GFP photoactivation to label individual KCs in animals subjected to HU ablation. The different KC classes have different median claw numbers: γ KCs have ~7 claws, αβ cells have ~5 claws, and α'β' KCs have ~3 claws (*Caron et al., 2013*); we therefore used two different strategies to label individual KCs in animals subjected to HU ablation, both of which allow us to assign KC type. First, we expressed PA-GFP broadly in Kenyon cells using MB247-Gal4, and assigned KC type by imaging the axonal lobe innervation of photoactivated cells (*Figure 2—figure supplement 3*). Second, we expressed PA-GFP only in αβ or γ KCs using 58F02-Gal4 (*Figure 2I*) and 89B01-Gal4, respectively. Combining data from these two methods, we found no change in claw number on individual αβ KCs (*Figure 2J–K*) or γ KCs (*Figure 2—figure supplement 3*) as the KC population was reduced. From these experiments, we conclude that PNs reduce their bouton production on a cell-by-cell basis as

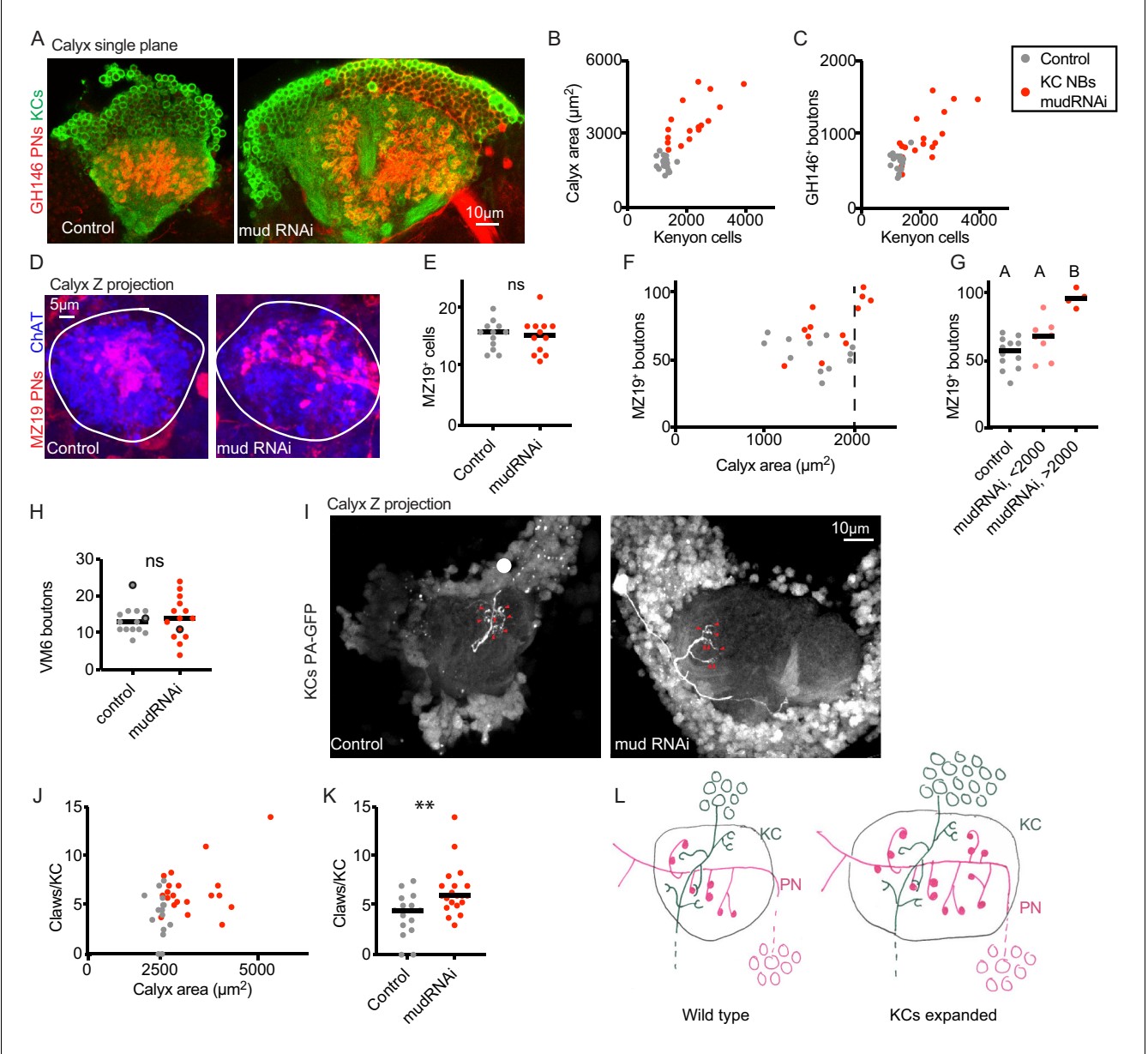

**Figure 3.** Increase in Kenyon cell number leads to increased projection neuron bouton complement. (A) Confocal slice of control calyx and calyx subjected to KC expansion by expression of *mud*-RNAi in KC neuroblasts via OK107-Gal4. GH146 labels PNs (red) and OK107 labels KCs (green). (B, C) Relationship between number of KCs and maximum calyx cross-sectional area (B) and PN bouton number (C) in control (gray) and KC neuroblast>mud RNAi hemispheres (red). (D) Maximum intensity projection of confocal stack of calyx bouton production by ~16 MZ19+ PNs (red) in control and KC neuroblast>mud RNAi. ChAT immunostaining (blue) highlights calyx extent. (E) Quantification of MZ19 cells in the two genotypes. Significance: unpaired t-test. Throughout this figure, black horizontal bars represent medians. (F, G) Relationship between maximum calyx cross-sectional area and MZ19 bouton production. Vertical dashed line indicates calyces larger than 2000 $\mu m^2$, observed only in the experimental genotype. Significance: one-way ANOVA. (H) Boutons of the VM6 PN, labeled by 71D09-LexA, in control (gray) or brains subjected to KC expansion via OK107>mudRNAi (red). Every hemisphere contained one VM6 neuron in both genotypes except the three circled datapoints, which had two VM6 PNs. Significance: unpaired t-test. (I) Maximum intensity projection of two-photon stack of single KCs highlighted by GFP photoactivation (C3PA driven by OK107) in control or OK107>mudRNAi calyces. Position of out-of-plane somata in control image indicated by white circle. Red arrowheads indicate scored claws. (J, K) Quantification of claws of KCs labeled as in (I). Y axis is average number of claws per KC as multiple KCs were photoactivated in some samples. Significance: unpaired t-test; here and throughout, *: p<0.05, **: p<0.01, ***: p<0.001, ****: p<0.0001. (L) Model of the effect of KC expansion on calyx development. KCs retain normal claw numbers while PNs increase their bouton numbers, increasing calyx size.

The online version of this article includes the following video and figure supplement(s) for figure 3:

**Figure supplement 1.** Claw number for α'β' KCs.

*Figure 3 continued on next page*

*Figure 3 continued*
**Figure 3—video 1.** Volume renderings of mushroom bodies shown in *Figure 3A*.
https://elifesciences.org/articles/52278#fig3video1

the KC population shrinks, while KC claw number is unaffected by the size of the KC population (model, *Figure 2L*).

## Olfactory projection neurons increase bouton repertoire as Kenyon cell number increases

In order to assess whether PNs can vary their presynaptic bouton number bidirectionally, we next created a method to specifically amplify the KC population. Recent work has suggested that ectopic or supernumerary populations of neurons can integrate into a variety of fruit fly neural circuits (*Meng et al., 2019*; *Pop et al., 2019*; *Prieto-Godino et al., 2019*; *Seroka and Doe, 2019*; *Shaw et al., 2018*), and previous studies identified a mutant, called *mushroom body defect* (*mud*) in which the mushroom body and other brain areas are dramatically expanded (*Guan et al., 2000*; *Prokop and Technau, 1994*). Mud/NuMA is a spindle orientation protein that ensures neuroblasts divide asymmetrically to produce one neuroblast and one differentiating neuron or neural progenitor (*Siller et al., 2006*). In *mud* mutants, neuroblasts occasionally divide symmetrically to produce two neuroblasts, amplifying the progeny of that neuroblast. To restrict amplification to KCs, we drove UAS-*mud*-RNAi (BL 35044) using OK107-Gal4, which is expressed in KC neuroblasts as well as in KCs (*Liu et al., 2015*). Importantly, in our RNAseq data from 45 hr APF and adult animals, we observe no expression of *mud* in PNs, KCs, or other brain cells, suggesting *mud* is not expressed in differentiating neurons (EJ Clowney, unpublished).

As OK107 labels mature KCs as well as KC neuroblasts, we included UAS-CD8GFP in these animals to count KCs. We observed potent amplification of KCs that was variable across animals, sometimes more than doubling KC number (*Figure 3A,B*). We measured calyx size, which expanded dramatically, and used GH146-driven fluorescence to count PN bouton number, which doubled in mushroom bodies with the largest KC expansions (*Figure 3C*, *Figure 3—video 1*). These results suggest that just as PNs scale down their boutons if the KC population is reduced, PNs scale up their boutons when the KC population is expanded.

To test whether PNs increased boutons on a cell-by-cell basis in these animals, we labeled VM6 using 71D09-LexA, and PNs innervating DA1, VA1d, and DC3 using MZ19-QF. First, we observed no change in VM6 or MZ19$^+$ cell numbers in brains with KC expansions (*Figure 3E,H*), suggesting that PN numbers do not increase as KC number increases. We observed no change in VM6 bouton numbers, perhaps because the VM6 PN already has a relatively large number of boutons (*Figure 3H*). In contrast, MZ19 cells increased their number of presynaptic boutons as the KC population expanded, from an average of 3.5 boutons per cells in the control genotype to an average of 4.75 boutons per cell in the KC-expansion genotype. In brains with measurably enlarged calyces, indicative of KC population expansion, MZ19$^+$ boutons doubled (*Figure 3D,F,G*).

To rule out reductions in the number of claws per KC as the KC population expands, we expressed photoactivatable GFP under the control of OK107 in animals with UAS-*mud*-RNAi or control and subjected individual or small groups of KCs to photoactivation. We scored the number of claws per labeled KC in each calyx. As above, calyx size predicted KC number in these animals, so we used calyx size as a proxy for KC number. Rather than a decrease in claw number per KC in animals with expanded KC complements, we found a slight increase in claw number, from a median of 5 to a median of 6 claws. As different types of KCs produce different characteristic numbers of claws, this could represent a change in claw production, a change in the distribution of KC types in these animals, or bias in which KC types we labeled in expanded versus control brains. While we were unable to identify LexA lines to specifically label αβ or γ KCs in OK107>mudRNAi brains, we found a LexA line, R41C07, that labeled α'β' KCs, and targeted these specifically for photoactivation (*Figure 3—figure supplement 1*). In these KCs of matched type, we found no change in KC claw number per cell, though we cannot rule out subtle differences. Together, these experiments suggest that in animals with an amplified KC complement, PNs dramatically increase their bouton production

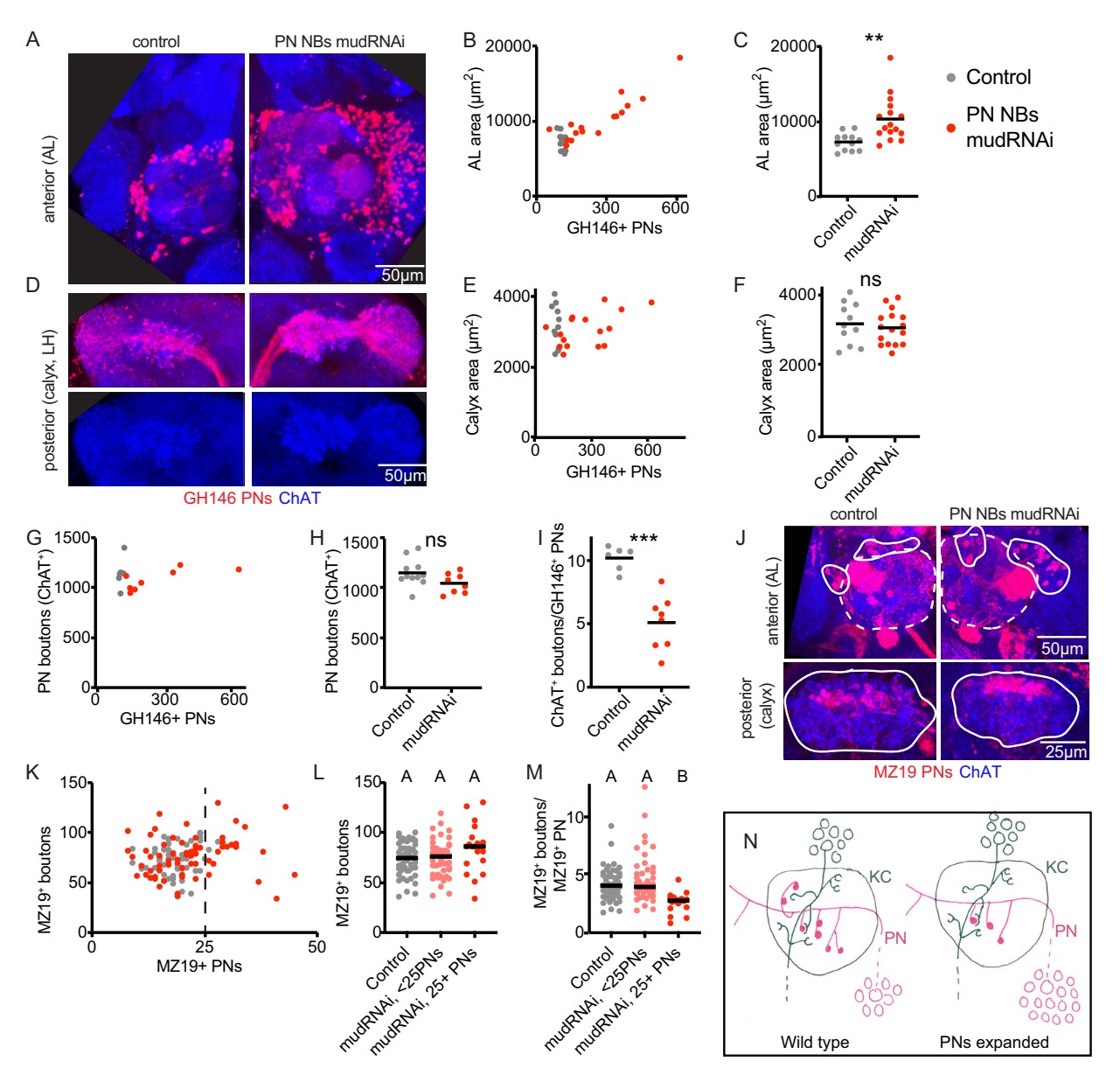

**Figure 4.** Individual projection neurons scale boutons down as projection neuron population increases. (**A**) Maximum intensity projection of confocal stack of anterior of the brain of control or animal subjected to PN expansion by expression of *mud*-RNAi in PN neuroblasts via 44F03-Gal4. GH146 labels PNs (red) and ChAT staining highlights cholinergic neurons. (**B, C**) Relationship between PN number (measured by counting GH146[+] cells, about 60% of the total PNs) and antennal lobe maximum cross-sectional area in control (gray) or PN neuroblast>mudRNAi hemispheres (red). Significance: unpaired t-test. Throughout this figure, black horizontal bars represent medians. (**D**) Maximum intensity projection of confocal stacks of the posteriors of the same brains shown in A. Excess PNs can be seen to project to the calyx, due to thickening of the axonal tract passing on top of the calyx (red), and produce increased innervation density in the lateral horn. (**E–I**) Relationship between PN number and maximum cross-sectional area of the calyx (**E**, **F**), PN bouton number (**G, H**), and ratio of ChAT[+] boutons to GH146[+] PNs (**I**) in control and PN neuroblast>mudRNAi brains. Significance: unpaired t-test. (**J**) Maximum-intensity projections of confocal stacks of antennal lobes (top) and calyces (bottom) of control and PN neuroblast>mudRNAi brains. MZ19[+] PNs labeled red. Dashed outlines on anterior show antennal lobe extent; solid outline on anterior, groups of PN somata; solid outlines on posterior, calyx extent. (**K–M**) Quantification of MZ19[+] boutons in control and PN neuroblast>mudRNAi brains. Dashed vertical line indicates 25 MZ19[+] PNs, as controls seldom exceed this number. Significance: one-way ANOVA. (**N**) Model of the effect of increasing PN numbers on calyx development. As PNs increase, PNs reduce bouton production so that the overall number of boutons, and calyx size, stays similar to wild type.

The online version of this article includes the following figure supplement(s) for figure 4:

**Figure supplement 1.** Additional quantification related to Figure 4.

to match the KC population, while KCs may moderately increase, and do not reduce, their individual claw production.

## Individual projection neurons decrease bouton number as the projection neuron population expands

We have shown that olfactory PNs bidirectionally vary their presynaptic boutons to suit populations of KCs derived from 0 to 8 KC neuroblasts. To ask whether PNs can also vary their bouton production in response to changes in their own numbers, or in response to changes in the ratio between PNs and KCs that do not affect KC clone number, we identified methods to vary the PN population. We found that just as we could expand the KC population by knocking down *mud* in KC neuroblasts, we could expand the PN population by knocking down *mud* in PN neuroblasts. This strategy produced GH146-labeled PN complements of up to 600 cells, as compared to ~120 GH146-labeled PNs we counted in controls. While the PN neuroblast Gal4 we used, 44F03, can label both lNB/BAlc and adNB/BAmv3 progeny (*Awasaki et al., 2014*), we found that ventrolateral PNs were much more susceptible to expansion than anterodorsal PNs.

Supernumerary PNs innervated the antennal lobe, causing a massive expansion of antennal lobe area (*Figure 4A,C*). The normal glomerular map in the AL appeared distorted, but glomerular divisions were still present. These PNs also projected to the calyx, observed by thickening of the PN axon tract crossing the calyx (red, *Figure 4D*), and increased the density of PN innervation of the lateral horn (*Figure 4D*), though not lateral horn area (*Figure 4—figure supplement 1*). However, there was no change in calyx area or total bouton number in these animals compared to wild type (*Figure 4E–H*). At a population level, the average number of boutons per PN was half that in controls (*Figure 4I*).

To ask how small groups of PNs adjusted to PN expansion, we again used MZ19-QF, which labels 15–20 PNs in controls (*Figure 4J,K*). We counted MZ19+ cells and boutons in control versus animals of the expansion genotype (*Figure 4K*). As the expansion was highly variable and we rarely observed more than 25 MZ19+ cells in controls, we divided the expanded genotype into hemispheres with <25 PNs or ≥25 PNs. There was a modest but insignificant increase in MZ19+ boutons in the ≥25 PNs groups (*Figure 4L*), attributable to a strong and significant reduction of average number of boutons per PN (*Figure 4M*). Thus PNs scale down their individual bouton repertoire to compensate for increases in the size of the PN population. Like MZ19 PNs, the VM6 PN also maintained a stable number of total boutons as the PN population expanded (*Figure 4—figure supplement 1*). This was possible because the VM6 cell(s) had similar numbers of total boutons as the VM6 population expanded.

## Projection neuron bouton expansion partially compensates for severe reductions in the projection neuron population

In order to reduce the PN repertoire independent of KC number, we identified two Gal4 lines, VT033006 and VT033008, that label large complements of uniglomerular PNs in the adult (*Figure 5A*). We drove diphtheria toxin A (*Berdnik et al., 2006*) and fluorescent reporters under control of these lines in animals where additional PNs were labeled by GH146-LexA (*Figure 5B,G*). In each case, DTA eliminated all Gal4+ cells as well as additional GH146+ PNs, suggesting that these lines drive DTA expression in broader PN populations at earlier stages of development. Driving DTA under control of VT033008 reduced the total PNs labeled by VT033008 and/or GH146 by nearly half (*Figure 5C*), while VT033006>DTA reduced PNs labeled by VT03006 and/or GH146 by >90% (*Figure 5H*). PN labeling by VT033006 and VT033008, and cell loss when used to drive DTA, were already evident in third instar larvae, suggesting PNs are lost well before pupal calyx wiring (*Figure 5—figure supplement 1*). While PN dendrites have been shown to occupy glomerular territories in pupal stages before OSNs innervate the antennal lobe (*Jefferis et al., 2004*), we found that antennal lobes lacking the vast majority of uniglomerular PNs retained glomerular divisions. This suggests that these PNs may not be required for the formation of glomerular divisions in the adult antennal lobe.

We measured maximum calyx cross-sectional area and bouton number across these conditions and found that in each case, the effect on the calyx was measurable but much less severe than the reduction in PN number. For VT033008>DTA, loss of nearly half the PNs yielded no change in calyx

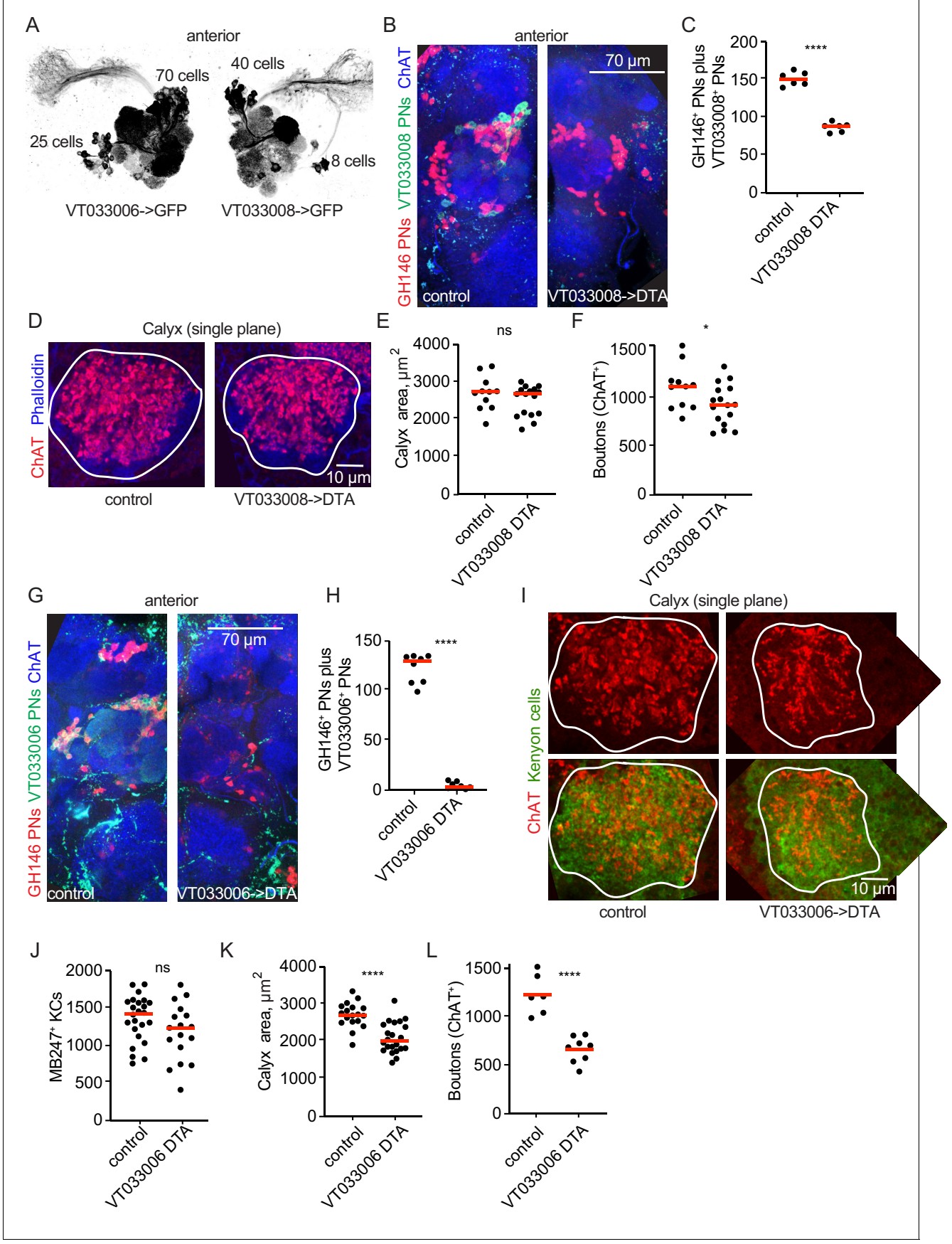

**Figure 5.** Projection neurons partially compensate for decreases in projection neuron population. (**A**) Maximum-intensity projections of two-photon stacks of brains with VT033006 (left) and VT033008 (right) driving GFP. (**B**) Maximum intensity projection of confocal stack of antennal lobes of control (left) and VT033008>DTA (right) brains. VT033008$^+$ cells shown in green, GH146$^+$ PNs, which partially overlap with VT033008$^+$ PNs, shown in red. (**C**) Quantitation of PNs positive for GH146, VT033008, or both in control and VT033008>DTA brains. All VT033008$^+$ cells are ablated in this genotype. Significance: unpaired t-test. Throughout this figure, red horizonal bars represent medians. (**D**) Single confocal slice of control or VT033008>DTA calyx. Phalloidin signal shown in blue and ChAT$^+$ boutons shown in red. (**E, F**) Quantification of calyx maximum cross-sectional area and bouton number in control and VT033008>DTA animals. Significance: unpaired t-test. (**G**) Maximum intensity projection of confocal stack of antennal lobes of control (left) and VT033006>DTA (right) brains. VT033006$^+$ cells shown in green, GH146$^+$ PNs, which partially overlap with VT033006$^+$ PNs, shown in red. Green signals outside the antennal lobe in both conditions are autofluorescence. (**H**) Quantitation of PNs positive for GH146, VT033006, or both in control and VT033006>DTA brains. All VT033006$^+$ cells are ablated in this genotype as well as most GH146$^+$ PNs, even those that are not VT033006$^+$ in the adult. Examination of ChAT$^+$ neurites entering the calyx in VT033006>DTA brains suggests that up to 10 PNs may be labeled by neither VT033006 nor GH146, and that some of these may survive ablation. Significance: unpaired t-test. (**I**) Single confocal slice of control (left) or VT033006>DTA (right) calyx. KCs labeled using MB247-dsRed (shown green for continuity), ChAT$^+$ boutons shown in red. (**J–L**) Quantification of the number of MB247$^+$ KCs (**J**), calyx area (**K**), and ChAT$^+$ boutons (**L**) in control or VT033006>DTA brains. Significance: unpaired t-test.

The online version of this article includes the following figure supplement(s) for figure 5:

**Figure supplement 1.** VT033006 and VT033008 action in late larval stage.
**Figure supplement 2.** Quantification of Kenyon cells by flow cytometry.

area and a 20% reduction in overall bouton number (*Figure 5D–F*). Remarkably, while we estimate that only ~10 PNs remain in VT033006>DTA animals (~7% the normal complement, *Figure 5H*), the number of boutons in the calyx was reduced by only half (*Figure 5H,L*). This suggests that individual PNs could have expanded their bouton production by as much as 6-fold. While remaining PNs sometimes appeared to make exuberant collections of boutons (e.g. *Figure 6D*), distortion of the antennal lobes in these animals prevented us from assigning identities to these spared PNs and comparing bouton numbers to matched individual PNs in controls.

This severe loss of inputs could lead to KC death. We therefore labeled and counted KCs in VT033006>DTA animals with severe PN loss. While we cannot rule out a small reduction in KC complement, at least 80% of KCs remained in these brains. We also subjected control and VT033006>DTA brains to flow cytometry and found that PN ablation brains had at least 85% as many KCs as did control brains (*Figure 5—figure supplement 2*).

## Projection neuron insufficiency has little effect on Kenyon cell claw number

We expected that while KC claw number was largely invariant in the face of increases and decreases of the KC population, KCs would be forced to reduce their claw number when PN bouton production was reduced by 50%, as in VT033006>DTA animals. We therefore compared individual KCs in these animals versus controls. As different KC types produce different numbers of claws in wild type, we sought to simplify the analysis by labeling only a single KC type. To do this, we targeted Texas red dextran dye electroporation to the tips of the mushroom body α lobes, which are innervated by αβ KCs (*Figure 6A*). This allowed us to label 1–12 KCs per calyx. We could discern discrete dye-labeled claws and somata in animals with eight or fewer KCs labeled, and restricted our analyses to these samples (*Figure 6B–D*). For each hemisphere we counted labeled somata and claws, generating one average value, claws per labeled KC, per separable group of cells (*Figure 6F*). Surprisingly, despite 90% reduction in the number of PNs and 50% reduction in their boutons, we observed no reduction in claw number per KC.

Do these enduring claws receive inputs? While labeled KCs in VT033006>DTA brains sometimes made clear contacts with spared PNs, as in *Figure 6D,E*, in other samples we also observed KCs that avoided the one or few GH146-labeled, spared PNs (data not shown). Besides PNs, GH146 labels an inhibitory neuron, APL, that innervates the mushroom body lobes and calyx. APL is presynaptic to KCs in the calyx, but innervates portions of KCs that are not claws (*Lin et al., 2014*; *Zheng et al., 2018*). Labeled KCs in VT033006>DTA brains still formed contacts with APL (*Figure 6C*).

A recent study in the *Drosophila* larval body wall found that neurons manipulated to produce excess dendrites can obtain input from atypical presynaptic partners (*Tenedini et al., 2019*). In wild type animals, 10% of bouton inputs to Kenyon cells in the main (olfactory) calyx are from gustatory

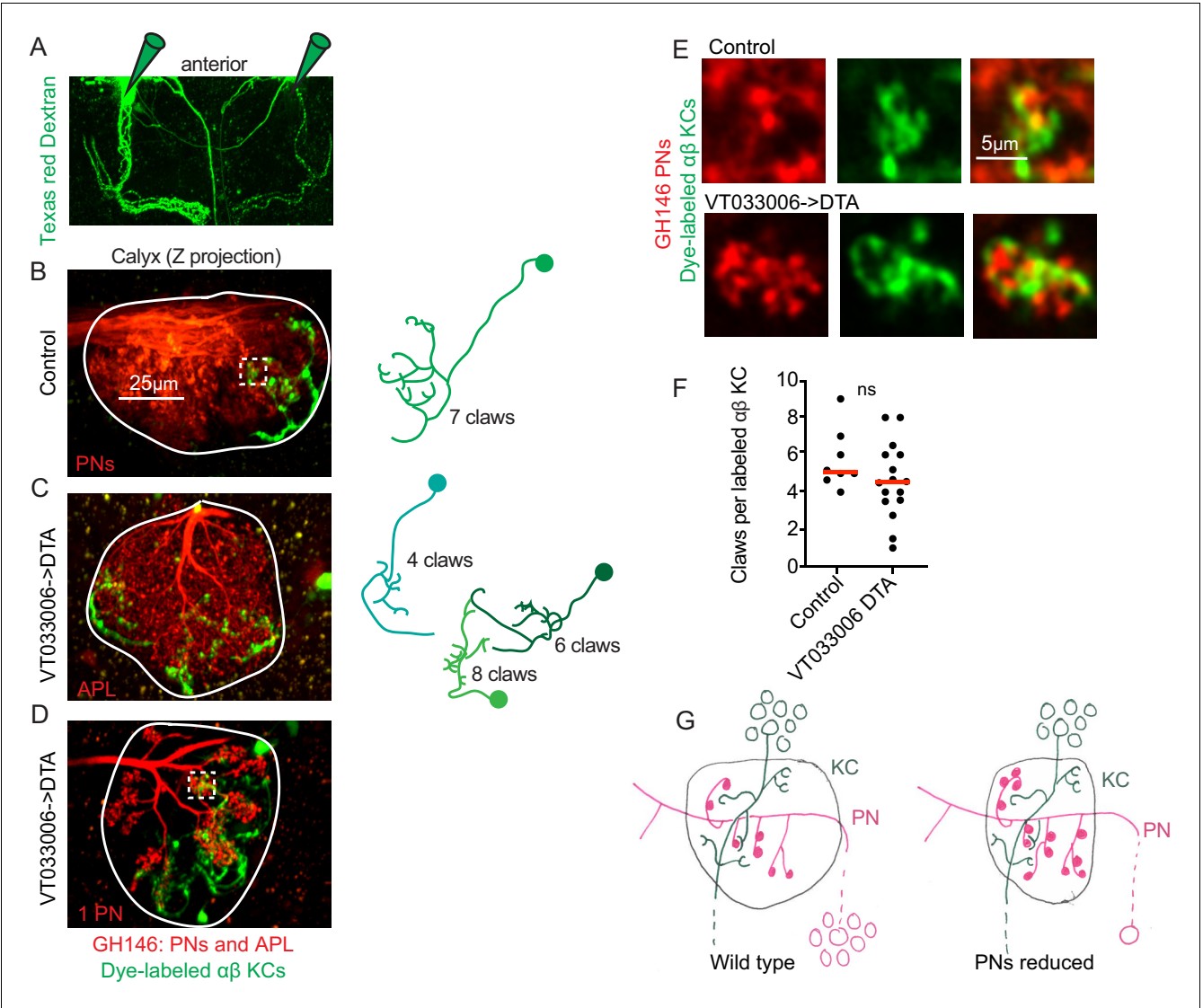

**Figure 6.** Kenyon cell claw number is unaffected by projection neuron bouton reduction. (**A**) Texas red dextran (colored green for continuity with green KCs in other figures) was electroporated into the tips of the mushroom body α lobes to allow specific targeting of αβ KCs. (**B–D**) Maximum intensity projections of two photon stacks of control (**B**) and VT033006>DTA (**C, D**) calyces. GH146, shown in red, labels PNs and APL. In control (**B**), APL is obscured by dense PN signal. In some VT033006>DTA hemispheres, as in (**C**), all GH146+ PNs were ablated, making APL signal apparent. In other VT033006>DTA hemispheres, as in (**D**), 1–2 GH146+ PNs remained. Texas red dextran dye diffusing from KC axons is shown in green. Diagrams show reconstructed anatomies of dye-labeled KCs. In (**B**), one KC with seven claws is labeled. In C, three KCs with 4, 6, and eight claws are labeled and can be spatially separated from one another. In D, 8 KCs are labeled and cannot be reconstructed. White boxes in (**B**) and (**D**) show regions magnified in (**E**). (**E**) Texas red dextran dye-labeled KC claws (green) wrapping GH146+ PN boutons (red) in control (top) and VT033006>DTA brains. (**F**) Quantification of claws per KC in control or VT033006>DTA samples. Each data point is the average number of claws per labeled KC for a spatially separated individual KC or small group of KCs. Significance: unpaired t-test. Red horizontal bars represent medians. (**G**) Model of the effect on calyx development of ablating large groups of PNs. Remaining PNs increase their production of boutons but cannot fully compensate for lost PNs, reducing the size of the calyx. KC claw numbers are similar to wild type; their source of innervation is unknown.

or thermal modalities (*Caron et al., 2013*; *Eichler et al., 2017*; *Frank et al., 2015*; *Kirkhart and Scott, 2015*; *Zheng et al., 2018*). It is possible that thermal or gustatory neurons projecting to the calyx could have also expanded their production of boutons to make up for the losses of olfactory PNs. We cannot yet determine whether the excess KC claws observed in this condition get inputs from PN boutons, from APL, or from some other source. Nevertheless, these data suggest that KCs are inflexible in their production of claws, even in the face of severe reduction of their typical, favored presynaptic partners.

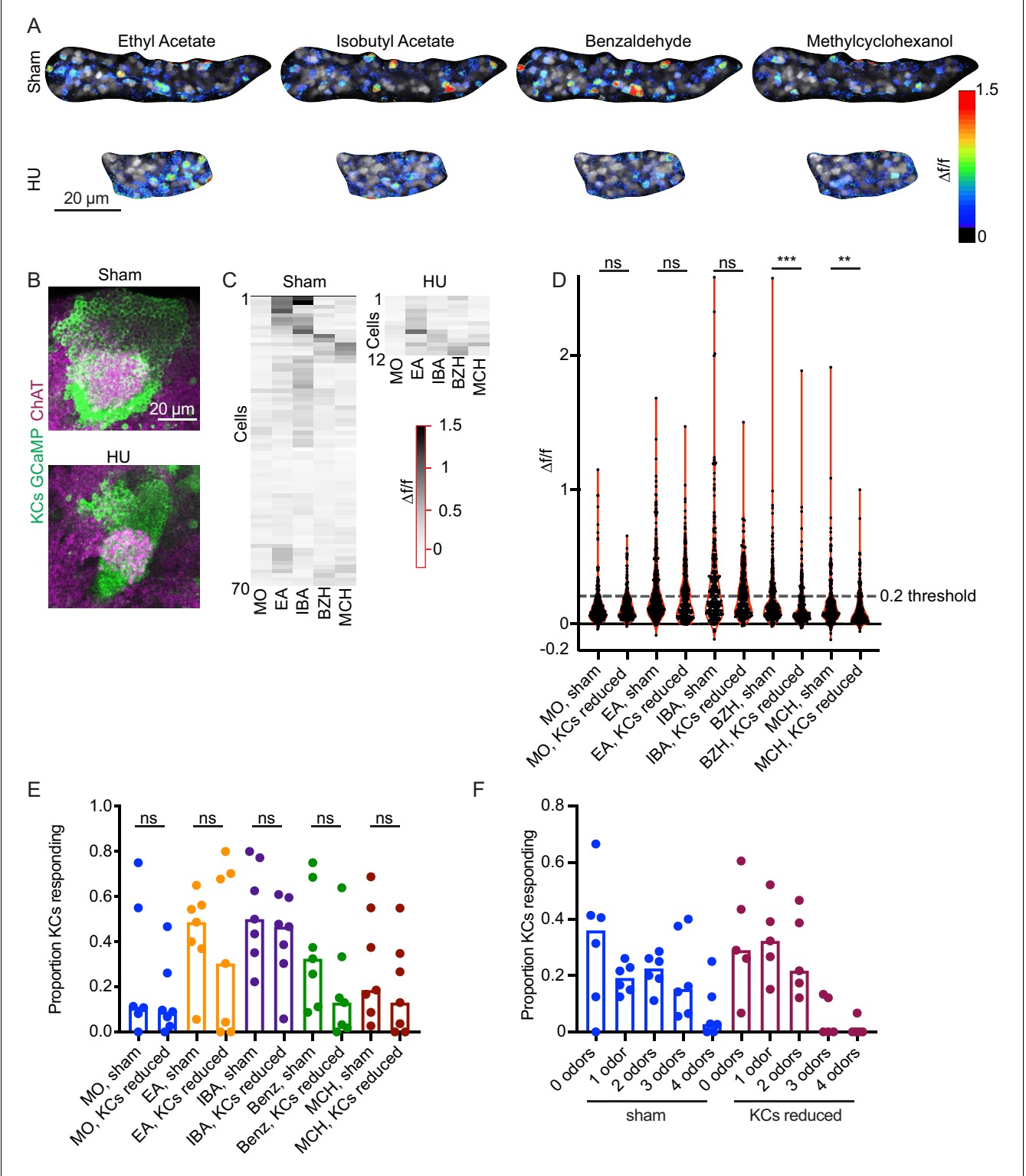

**Figure 7.** Odor responses in calyces with reduced KCs. (A) Example KC somatic odor responses in sham-treated hemisphere and HU-treated hemisphere with reduced KCs. (B) Post-imaging immunostaining showing calyces and KC somata of samples shown in (A). (C) Peak post-odor responses of individual KCs from samples shown in (A). Only cells that remained stable in their ROI for the whole stimulus panel are shown. Data was clustered by row such that cells with more similar odor responses are closer to one another. MO: Mineral oil (mechanosensory control), EA: Ethyl Acetate, IBA: Isobutyl Acetate, BZH: Benzaldehyde, MCH: Methylcyclohexanol. (D) Peak odor responses of all cells, aggregated from all analyzed samples. Dashed line indicates 20% Δf/f threshold. Significance: Mann-Whitney test. (E) Fraction of cells in each sample responding to each odor at

*Figure 7 continued on next page*

*Figure 7 continued*

20% Δf/f threshold. Significance: unpaired t-test. (F) For samples in which the same cells could be tracked across all odor presentations, fraction of cells responding to 0, 1, or multiple odors is shown. Bar plots in (E, F) show medians.

The online version of this article includes the following video, source data, and figure supplement(s) for figure 7:

**Source data 1.** Peak response data for all animals in Figure 7.
**Figure supplement 1.** Odor responses over time for cells shown in *Figure 7A–C*.
**Figure 7—video 1.** Full, motion-corrected functional imaging time series for one sham-treated calyx.
https://elifesciences.org/articles/52278#fig7video1
**Figure 7—video 2.** Full, motion-corrected functional imaging time series for one reduced-KC calyx.
https://elifesciences.org/articles/52278#fig7video2

## Developmental plasticity preserves sparse odor coding despite perturbations to cell populations

The disparate and numerous strategies we used above to vary the ratio between PNs and KCs all produced the same result, that PNs adjusted to changes in PN::KC ratio by altering their presynaptic boutons, while the distribution of KC claw numbers changed little even when perturbations to PNs were so severe as to prevent production of the normal bouton complement. This finding suggests that developmental mechanisms prioritize the sparseness of olfactory inputs to KCs, and might therefore preserve sparse odor coding and high-dimensional odor representations despite stark changes to the underlying circuit constituents. We therefore sought to ask how KC odor responses were affected by perturbations to the PN::KC ratio.

We used OK107 to drive GCaMP6s expression in all KCs, and subjected animals to HU just after larval hatching. We then imaged KC somatic odor responses in vivo in 24 adult HU-treated and 14 adult sham-treated animals. We observed robust odor responses in both conditions, with 8/28 sham-treated and 4/46 HU-treated calyces completely unresponsive. To identify HU-treated hemispheres with reduced KC complement, following functional imaging we imaged the anatomy of the calyx by two-photon or by immunostaining and confocal. Among ablated animals, we identified 14 of 46 imaged hemispheres in which calyces were clearly still present but of reduced size, which we designate as 'reduced-KC' calyces. We subjected these datasets and 17 responsive controls to motion correction using Suite2p (*Pachitariu et al., 2017*). After motion correction, seven sham and seven reduced KC calyces were sufficiently still to allow us to define ROIs for individual somata. For each cell, we calculated peak change in fluorescence following odor stimulus (*Figure 7—source data 1*). Example responses for sham-treated and reduced-KC calyces are shown in *Figure 7A, C*, *Figure 7—figure supplement 1*, and in *Figure 7—video 1, 2*.

In order to compare response magnitudes across conditions, we first pooled all cells from each condition and compared overall Δf/f in response to each odor (*Figure 7D*). The distributions were statistically indistinguishable between conditions for mineral oil (mechanosensory control), ethyl acetate, and isobutyl acetate, and significantly different for benzaldehyde and methylcyclohexanol. Using these aggregate responses, we set a threshold for 'responsive' cells of 20% increase in fluorescence over baseline, as we observed bifurcation of the cellwise response distribution at this cutoff. Using these criteria, a median of ~50% of cells from sham calyces responded to ethyl acetate and isobutyl acetate, and ~20–30% of cells to benzaldehyde and methylcyclohexanol (*Figure 7E*). These response rates are higher than has been reported with GCaMP3 and GCaMP1 (*Honegger et al., 2011*; *Wang et al., 2004*), but correspond with spiking response rates observed electrophysiologically (*Murthy et al., 2008*). We observed variation in odor responses in both control and reduced-KC calyces; interindividual variation is expected given the stochastic innervation of KCs by PNs.

We next compared the proportions of KCs per calyx responding to each odor between sham and reduced-KC calyces. These distributions were statistically indistinguishable (*Figure 7E*). Finally, we asked how many odors each cell responded to (*Figure 7F*). In each condition, ~60% of cells responded to 0 or one odors. Together, these findings suggest that the developmental plasticity mechanisms operating to set the density of olfactory inputs to KCs preserve qualitatively sparse KC odor responses when the KC population is experimentally reduced. Further experiments will be required to assess quantitative effects of KC reduction on responses to odors that typically stimulate fewer cells, like benzaldehyde and methylcyclohexanol.

## Discussion

Cerebellum-like architectures are found repeatedly in phylogenetically distinct organisms, suggesting that they have broad evolutionary utility. Yet the developmental production of sparseness, a key wiring feature allowing high-dimensional sensory representations, is not understood. Here, we have begun to investigate the development of sparse connectivity in the cerebellum-like associative learning center of insects, the mushroom body. By varying the ratio between presynaptic olfactory PNs and postsynaptic KCs, we find that connectivity density is set by KCs: KC claw number changes little across a wide range of PN::KC ratios, KC number predicts PN bouton number, and PNs exhibit wide variety in bouton number to satisfy KC demands. As described below, we expect that this strategy for generating connectivity density would preserve sparseness in KC inputs across developmental time and upon evolutionary changes in cell repertoires and thus maintain the olfactory coding function of the mushroom body over a range of conditions.

### Implications for development: Different projection neuron to Kenyon cell ratios across developmental stages

For many animals, brain development continues while juveniles are already living freely–searching for food, navigating, and learning about the environment. Developmental transitions in brain structure are particularly stark in holometabolous insects, who build the brain twice. In *D. mel*, neurons generated during embryonic and early larval stages wire the larval nervous system. These circuits support larval behaviors, while neural stem cells continue to generate additional neurons that will be used to build the adult circuit. In keeping with this, the ratio between PNs and KCs in the larval olfactory circuit is starkly different from the adult: 21 embryonically-born PNs wire to early-born KCs to construct the larval mushroom body (*Eichler et al., 2017*; *Masuda-Nakagawa et al., 2005*; *Ramaekers et al., 2005*). Connections among these populations dissolve in early pupae, and are then re-wired during pupal development (*Lee et al., 1999*; *Marin et al., 2005*), joined by ~100 more larvally-born PNs, and >1000 more KCs per hemisphere that continue to be born until immediately before adult eclosion.

The 21 PNs in the early larva connect to ~75 KCs, a 1:3 ratio, while in the adult, ~150 PNs connect to ~2000 KCs, a 1:10 ratio. (*Eichler et al., 2017*; *Zheng et al., 2018*). We found that unlike cells in many other systems, including the vertebrate cerebellum, PNs and KCs did not rely on each other for survival signals (*Fan et al., 2001*). This may be due to the constantly changing ratio between these cell types across developmental time. Instead, setting connectivity density cell-autonomously in KCs could allow KCs to obtain the appropriate number of inputs at the different life stages of the animal, when cellular constituents are very different from one another. Similarly, while PN neurogenesis ceases well before PNs and KCs begin to contact one another in the pupa, we estimate that ~10% of KCs are born after PN:KC synapse has already initiated (*Muthukumar et al., 2014*). Strict, cell-autonomous dendrite structuring and flexible PN bouton production could together ensure that late-born KCs obtain the inputs appropriate to support coding.

### Implications for coding: Balancing projection neuron representations across the calyx

Olfactory PNs of different types are specified in a predictable temporal order, have characteristic numbers of boutons, and overlap in their innervation of the calyx (*Lai et al., 2008*; *Yu et al., 2010*; *Zheng et al., 2018*). Differences in bouton number across different PNs allow different odor channels to be differentially represented in the calyx and in KC odor responses (*Caron et al., 2013*; *Honegger et al., 2011*; *Murthy et al., 2008*). Several classes of PNs also differ in number between the sexes (*Grabe et al., 2016*). While PNs changed their individual bouton repertoires in response to changes in cell repertoires, we found that to some extent, the representation level of different PNs in the calyx was preserved. For example, in *Figure 2F* and *Figure 2—figure supplement 2*, we show the effect of reducing KC number on bouton production by the VM6 PN and 42D01 PNs. While each population of PNs reduced individual bouton number in this condition, they retained their typical relative representation. The VM6 PN reduced its boutons from 10 to 5, while the 42D01 PNs decreased their boutons from 4 to 2. Similarly, in *Figure 4*, we expanded the PN population by inducing ectopic PN neuroblast duplication. In these experiments, we mainly observed amplification of the ventrolateral clone. We found that individual anterodorsal VM6 cells did not scale down their

boutons when the ventrolateral PN clone expanded, but only when VM6 itself was duplicated. Again, this could maintain the relative wild type representations of different odor channels in the calyx. A recent analysis suggests that the spontaneous activity of different ORs correlates with number of boutons representing that odor channel in the calyx (*Kennedy, 2019*). One possible model for how PNs scale to KC numbers while maintaining their relative representations in the calyx is thus that KC number limits total bouton number across all PNs, while allocation of these boutons to individual PNs is determined by activity-based competition among PN types.

## Implications for coding: Maximizing dimensionality of odor representations

Qualitative aspects of sparse coding in the mushroom body appear robust to severe perturbations to the circuit. Alternative developmental compensatory mechanisms would be much less likely to preserve sparse coding. For example, we increased the ratio of PNs to KCs in two ways, by increasing the number of PNs and by decreasing the number of KCs. In both cases, PNs dialed down their bouton number, making 25–50% of the boutons they make in wild type. This allowed the KCs to receive their typical number of inputs. If in contrast bouton number was rigid and claw number flexible, in these cases KCs would have expanded their claw production 2–4 fold to innervate all the incoming PN boutons. Individual KCs with for example 20 instead of 5 claws would receive input from ~40% of glomeruli, increasing the overlap in tuning across different KCs and degrading the ability of the network to distinguish different stimuli from one another (*Litwin-Kumar et al., 2017*).

In two other cases, we increased the ratio of KCs to PNs, by increasing the number of KCs and by decreasing the number of PNs. Again, KCs retained their typical claw number. If instead PNs had maintained a static production of boutons while KCs had adjusted their claw production, KCs would receive very few inputs. While increasing the number of inputs per KC is theorized to reduce dimensionality of odor responses by making different KCs more similar to one another, decreasing the number of inputs per KC is theorized to reduce dimensionality by reducing the number of different possible KC input combinations (*Litwin-Kumar et al., 2017*). That this sweet spot maximizing dimensionality, ~5 inputs per cell, is programmed into KC identity testifies to the evolutionary importance of maintaining connectivity density in associative brain centers that rely on combinatorial coding.

## Implications for evolution: The potential for mushroom body function despite perturbations

The olfactory receptors are the largest insect gene family and have been subject to frequent and extreme gains and losses in many clades. Similarly, brain centers devoted to learning are radically different across species, as exemplified by the diversity in KC repertoire across arthropods (*Strausfeld et al., 2009*). In order to acquire a novel olfactory circuit, many different evolutionary changes are required: A new receptor must evolve, an OSN type that uniquely expresses the receptor needs to arise, that OSN needs to synapse onto PNs, and a new PN type and new glomerulus must arise. For these events to accrue over time, each individual change must be compatible with continued circuit function and olfactory behavior. While development of a dedicated circuit that assigns an innate meaning to a newly-detectable odor would require many further changes, the signal could add to fitness immediately through representation in the mushroom body.

We have described two mechanisms of developmental robustness that maintain coherent mushroom body wiring in the face of a broad range of phenotypic alterations. First, we observe that olfactory PNs can adjust to gain and loss of PNs while maintaining the balance of odor channel representations in the calyx. Plastic development of PN presynaptic sites that makes room for additional players in the repertoire would allow immediate access of evolutionarily expanded PNs to the calyx and the use of their signals for olfactory learning, thus making time for the evolution of hardwired circuits for innate interpretations. Second, we show here that developmental programs wiring the calyx can accommodate variation in KC number from at least ¼ to 2-fold the endogenous complement. Again, this flexibility could support continued MB function on the path to the evolution of mushroom body innovations. Future experiments will ask how KC claw number is developmentally programmed, and what mechanisms operate in olfactory PNs to allow them to tailor bouton production to the KC repertoire.

# Materials and methods

**Key resources table**

| Reagent type (species) or resource | Designation | Source or reference | Identifiers | Additional information |
|---|---|---|---|---|
| Gene (*Drosophila melanogaster*) | mud | | FLYB: FBgn0002873 | |
| Genetic reagent (*Drosophila melanogaster*) | GH146Gal4 (II) | Bloomington Drosophila Stock Center | BDSC 30026 | (*Stocker et al., 1997*) |
| Genetic reagent (*Drosophila melanogaster*) | 10XUAS-IVS-myr::tdTomato (attp40) | Bloomington Drosophila Stock Center | BCSC 32222 | (*Pfeiffer et al., 2012*) |
| Genetic reagent (*Drosophila melanogaster*) | MB247-LexAVP16 (III) | Scott Waddell, University of Oxford | | (*Pitman et al., 2011*) |
| Genetic reagent (*Drosophila melanogaster*) | LexAop-CD2-GFP (III) | Bloomington Drosophila Stock Center | BDSC 66544 | (*Lai and Lee, 2006*) |
| Genetic reagent (*Drosophila melanogaster*) | GH146LexA (II) | Tzumin Lee, Janelia Farms Research Campus | | (*Lai et al., 2008*) |
| Genetic reagent (*Drosophila melanogaster*) | lexAop-mRFP-nls (II) | Bloomington Drosophila Stock Center | BDSC 29956 | Gunter Merdes |
| Genetic reagent (*Drosophila melanogaster*) | 58F02-Gal4 (attp2) | Bloomington Drosophila Stock Center | BDSC 39186 | (*Jenett et al., 2012*) |
| Genetic reagent (*Drosophila melanogaster*) | UAS-GCaMP6s (attp40) | Bloomington Drosophila Stock Center | BDSC 42746 | (*Akerboom et al., 2012*) |
| Genetic reagent (*Drosophila melanogaster*) | UAS-GCaMP6s (VK0005) | Bloomington Drosophila Stock Center | BDSC 42749 | (*Akerboom et al., 2012*) |
| Genetic reagent (*Drosophila melanogaster*) | MB247-DsRed (II) | André Fiala | | (*Riemensperger et al., 2005*) |
| Genetic reagent (*Drosophila melanogaster*) | 71D09-Gal4 (attp2) | Bloomington Drosophila Stock Center | BDSC 39584 | (*Jenett et al., 2012*) |
| Genetic reagent (*Drosophila melanogaster*) | UAS-CD8GFP (III) | Bloomington Drosophila Stock Center | BDSC 5130 | (*Lee and Luo, 1999*) |
| Genetic reagent (*Drosophila melanogaster*) | 42D01-Gal4 (attp2) | Bloomington Drosophila Stock Center | BDSC 50152 | (*Jenett et al., 2012*) |
| Genetic reagent (*Drosophila melanogaster*) | UAS-C3PA (II) | Vanessa Ruta, Rockefeller University | | (*Ruta et al., 2010*) |
| Genetic reagent (*Drosophila melanogaster*) | UAS-C3PA (III) | Vanessa Ruta, Rockefeller University | | (*Ruta et al., 2010*) |
| Genetic reagent (*Drosophila melanogaster*) | 60A10-LexA (attp40) | Bloomington Drosophila Stock Center | BDSC 52821 | Pfeiffer, Rubin LexA Collection |
| Genetic reagent (*Drosophila melanogaster*) | LexAop-tdTomato.Myr (su(Hw)attp5) | Bloomington Drosophila Stock Center | BDSC 56142 | (*Chen et al., 2014*) |
| Genetic reagent (*Drosophila melanogaster*) | 89B01-Gal4 (attp2) | Bloomington Drosophila Stock Center | BDSC 40541 | (*Jenett et al., 2012*) |
| Genetic reagent (*Drosophila melanogaster*) | 13XLexAop2-IVS-Syn21-mC3PA-GFP-p10 (attp2) | Gerry Rubin, Janelia Farms Research Campus | | (*Pfeiffer et al., 2012*) |
| Genetic reagent (*Drosophila melanogaster*) | GH146QF, QUAS-mtdTomato-3xHA (II) | Bloomington Drosophila Stock Center | BDSC 30037 | (*Potter et al., 2010*) |
| Genetic reagent (*Drosophila melanogaster*) | UAS-mud-RNAi (attp2) | Bloomington Drosophila Stock Center | BDSC 35044 | (*Perkins et al., 2015*) |
| Genetic reagent (*Drosophila melanogaster*) | OK107Gal4 (IV) | Bloomington Drosophila Stock Center | BDSC 854 | (*Connolly et al., 1996*) |
| Genetic reagent (*Drosophila melanogaster*) | P(caryP) (attp2) | Bloomington Drosophila Stock Center | BDSC 36303 | (*Perkins et al., 2015*) |
| Genetic reagent (*Drosophila melanogaster*) | P(caryP) (attp40) | Bloomington Drosophila Stock Center | BDSC 36304 | (*Perkins et al., 2015*) |

*Continued on next page*

*Continued*

| Reagent type (species) or resource | Designation | Source or reference | Identifiers | Additional information |
|---|---|---|---|---|
| Genetic reagent (*Drosophila melanogaster*) | MZ19$^{QF}$ (II) | Bloomington Drosophila Stock Center | BDSC 41573 | (*Hong et al., 2012*) |
| Genetic reagent (*Drosophila melanogaster*) | QUAS-mtdTomato-3xHA (II) | Bloomington Drosophila Stock Center | BDSC 30004 | (*Potter et al., 2010*) |
| Genetic reagent (*Drosophila melanogaster*) | 41C07-LexA (attp40) | Bloomington Drosophila Stock Center | BDSC 54791 | Pfeiffer, Rubin LexA Collection |
| Genetic reagent (*Drosophila melanogaster*) | 89B01-LexA (attp40) | Bloomington Drosophila Stock Center | BDSC 54380 | Pfeiffer, Rubin LexA Collection |
| Genetic reagent (*Drosophila melanogaster*) | 71D09-LexA (attp40) | Bloomington Drosophila Stock Center | BDSC 54931 | Pfeiffer, Rubin LexA Collection |
| Genetic reagent (*Drosophila melanogaster*) | LexAop-SPA-T2A-SPA (II) | n/a | | (*Clowney et al., 2015*) |
| Genetic reagent (*Drosophila melanogaster*) | 44F03-Gal4 (attp2) | Tzumin Lee, Janelia Farms Research Campus | | (*Awasaki et al., 2014*) |
| Genetic reagent (*Drosophila melanogaster*) | VT033006-Gal4 (attp2) | Yoshi Aso, Janelia Farms Research Campus | | (*Kvon et al., 2014*) |
| Genetic reagent (*Drosophila melanogaster*) | VT033008-Gal4 (attp2) | Yoshi Aso, Janelia Farms Research Campus | | (*Kvon et al., 2014*) |
| Genetic reagent (*Drosophila melanogaster*) | UAS-DTA (UAS-Cbbeta\DT-A.I) [18] (II) | Bloomington Drosophila Stock Center | BDSC 25039 | (*Han et al., 2000*) |
| Antibody | ChAT (mouse monoclonal) | Developmental Studies Hybridoma Bank | 9E10 | (1:200) |
| Antibody | DsRed (rabbit polyclonal) | Clontech | 632496 | (1:500-1:1000) |
| Antibody | GFP (sheep polyclonal) | Bio-Rad | 4745–1051 | (1:250-1:1000) |
| Antibody | GFP (chicken polyclonal) | Gift from Dawen Cai | n/a | (1:5000) |
| Antibody | Alexa 488 anti-mouse (goat polyclonal) | Fisher | A11029 | (1:500) |
| Antibody | Alexa 647 anti-mouse (goat polyclonal) | Fisher | A21236 | (1:500) |
| Antibody | Alexa 488 anti-sheep (donkey polyclonal) | Fisher | A11015 | (1:500) |
| Antibody | Alexa 488 anti-chicken (goat polyclonal) | Fisher | A11039 | (1:500) |
| Antibody | Alexa 568 anti-rabbit (goat polyclonal) | Fisher | A11036 | (1:500) |
| Chemical compound, drug | Texas Red Dextran | Fisher | D3328 | |
| Chemical compound, drug | Alexa 568 Phalloidin | Fisher | A12380 | |
| Chemical compound, drug | Schneider's medium | Sigma | S0146 | |
| Chemical compound, drug | Paraformaldehyde | EMS | 15710 | |
| Chemical compound, drug | Hydroxyurea | Sigma | H8627 | |
| Chemical compound, drug | Benzaldehyde | Sigma | 418099 | |

*Continued on next page*

*Continued*

| Reagent type (species) or resource | Designation | Source or reference | Identifiers | Additional information |
|---|---|---|---|---|
| Chemical compound, drug | Isobutyl acetate | Sigma | 537470 | |
| Chemical compound, drug | Ethyl acetate | Sigma | 270989 | |
| Chemical compound, drug | Methylcyclohexanol | Sigma | 153095 | |
| Peptide, recombinant protein | Collagenase | Sigma | C0130 | |
| Software, algorithm | Suite2p | (*Pachitariu et al., 2017*) | | |
| Other | Duck EZ Start Packaging tape | Office Depot | 511879 | |
| Other | UV Glue | Loctite | 3106 | |
| Other | JAXMAN 365 nm Flashlight | Amazon | B077GPXBK1 | |
| Other | Electra Waxer | Almore | 66000 | |
| Other | Gulfwax Paraffin | Grocery store | C0130 | |
| Other | hair | self | | |

## Flies

Flies were maintained on cornmeal-molasses ('R') food (Lab Express, Ann Arbor, MI) in a humidified incubator at 25C on a 12:12 light:dark cycle.

Genotypes were as follows:

| | |
|---|---|
| *Figure 1*, and *Figure 1—video 1* *Figure 1—figure supplement 1* | GH146$^{Gal4}$, 10XUAS-IVS-myr::tdTomato/ CyO; MB247-lexA, lexAop-CD2-GFP/TM2 |
| *Figure 1—figure supplement 2* | GH146$^{LexA}$, lexAop-mRFP-nls/P(caryP) attp40; VT033006-Gal4, UAS-CD8GFP/+ |
| *Figure 2A–E* | GH146$^{LexA}$, lexAop-mRFP-nls/CyO; 58F02-Gal4, UAS-GCaMP6s/TM2 |
| *Figure 2F–H* | MB247-DsRed/CyO; 71D09-Gal4, UAS-CD8GFP/" |
| *Figure 2I* | 60A10-LexA, LexAop-tdTomato.Myr/ CyO; 58F02-Gal4, UAS-C3PA/" |
| *Figure 2J,K* | GH146$^{Gal4}$, 10XUAS-IVS-myr::tdTomato/CyO; MB247-lexA, 13XLexAop2-IVS-Syn21-mC3PA-GFP-p10/" |
| | 60A10-LexA, LexAop-tdTomato.Myr/CyO; 58F02-Gal4, UAS-C3PA/" |
| *Figure 2—figure supplement 1* | GH146$^{LexA}$, lexAop-mRFP-nls/CyO; 58F02-Gal4, UAS-GCaMP6s/TM2 |
| *Figure 2—figure supplement 2* | MB247-DsRed/CyO; 42D01-Gal4, UAS-C3PA/" |
| *Figure 2—figure supplement 3A,B* | GH146$^{Gal4}$, 10XUAS-IVS-myr::tdTomato/CyO; MB247-lexA, 13XLexAop2-IVS-Syn21-mC3PA-GFP-p10/" |
| *Figure 2—figure supplement 3C,D* | GH146$^{Gal4}$, 10XUAS-IVS-myr::tdTomato/CyO; MB247-lexA, 13XLexAop2-IVS-Syn21-mC3PA-GFP-p10/" |
| | 60A10-LexA, LexAop-tdTomato.Myr/CyO; 89B01-Gal4, UAS-C3PA/" |
| *Figure 3A–C* | GH146$^{QF}$, QUAS-mtdTomato-3xHA /+; UAS-CD8GFP/UAS-mud-RNAi; OK107$^{Gal4}$/+ |
| | GH146$^{QF}$, QUAS-mtdTomato-3xHA /+; UAS-CD8GFP/P(caryP)attp2; OK107$^{Gal4}$/+ |
| *Figure 3D–G* | MZ19$^{QF}$, QUAS-mtdTomato-3xHA/Bl; UAS-mud-RNAi/TM2 or TM6B; OK107$^{Gal4}$/+ |

*Continued on next page*

| | |
|---|---|
| | MZ19<sup>QF</sup>, QUAS-mtdTomato-3xHA/Bl; P(caryP)attp2/TM2 or TM6B; OK107/+ |
| *Figure 3H* | 71D09-LexA, LexAop-tdTomato.Myr/Bl; UAS-mud-RNAi/TM2; OK107 <sup>Gal4</sup>/+ |
| | 71D09-LexA, LexAop-tdTomato.Myr/Bl; P(caryP)attp2/TM2; OK107 <sup>Gal4</sup>/+ |
| *Figure 3I* | UAS-C3PA/+; UAS-C3PA/UAS-mud-RNAi; OK107<sup>Gal4</sup>/+ |
| | UAS-C3PA/+; UAS-C3PA/P(caryP)attp2; OK107<sup>Gal4</sup>/+ |
| *Figure 3J,K* | UAS-C3PA/+; UAS-C3PA/UAS-mud-RNAi; OK107<sup>Gal4</sup>/+ |
| | UAS-C3PA/+; UAS-C3PA/P(caryP)attp2; OK107<sup>Gal4</sup>/+ |
| | UAS-C3PA/41C07-LexA, LexAop-tdTomato.Myr; UAS-C3PA/UAS-mud-RNAi; OK107<sup>Gal4</sup>/+ |
| | UAS-C3PA/41C07-LexA, LexAop-tdTomato.Myr; UAS-C3PA/P(caryP)attp2; OK107 <sup>Gal4</sup>/+ |
| | UAS-C3PA/89B01-LexA, LexAop-tdTomato.Myr; UAS-C3PA/UAS-mud-RNAi; OK107 <sup>Gal4</sup>/+ |
| | UAS-C3PA/89B01-LexA, LexAop-tdTomato.Myr; UAS-C3PA/P(caryP)attp2; OK107 <sup>Gal4</sup>/+ |
| *Figure 3—figure supplement 1* | 41C07-LexA, LexAop-tdTomato.Myr/UAS-C3PA; UAS-mud-RNAi 35044/UAS-C3PA; OK107 <sup>Gal4</sup>/+ |
| | 41C07-LexA, LexAop-tdTomato.Myr/UAS-C3PA; P(caryP)attp2/UAS-C3PA; OK107 <sup>Gal4</sup>/+ |
| *Figure 4A–I* | GH146<sup>LexA</sup>, LexAop-SPA-T2A-SPA/+; 44F03-Gal4/UAS-mud-RNAi |
| | GH146<sup>LexA</sup>, LexAop-SPA-T2A-SPA/+; 44F03-Gal4/P(caryP)attp2 |
| *Figure 4J–M* | GH146<sup>LexA</sup>, LexAop-SPA-T2A-SPA/MZ19<sup>QF</sup>, QUAS-mtd Tomato-3xHA; 44F03-Gal4/UAS-mud-RNAi |
| | GH146<sup>LexA</sup>, LexAop-SPA-T2A-SPA/MZ19<sup>QF</sup>, QUAS-mtd Tomato-3xHA; 44F03-Gal4/P(caryP)attp2 |
| | MZ19<sup>QF</sup>, QUAS-mtdTomato-3xHA/+; 44F03-Gal4/UAS-mud-RNAi |
| | MZ19<sup>QF</sup>, QUAS-mtdTomato-3xHA/+; 44F03-Gal4/P(caryP)attp2 |
| *Figure 4—figure supplement 1A,B* | GH146<sup>LexA</sup>, LexAop-SPA-T2A-SPA/+; 44F03-Gal4/UAS mud-RNAi |
| | GH146<sup>LexA</sup>, LexAop-SPA-T2A-SPA/+; 44F03-Gal4/P(caryP)attp2 |
| *Figure 4—figure supplement 1C–F* | 71D09-LexA, LexAop-tdTomato.Myr/+; 44F03-Gal4/UAS-mud-RNAi |
| | 71D09-LexA, LexAop-tdTomato.Myr/+; 44F03-Gal4/P(caryP)attp2 |
| *Figure 5A* | CyO/+; VT033006-Gal4/UAS-CD8GFP |
| | CyO/+; VT033008-Gal4/UAS-CD8GFP |
| *Figure 5B,C* | GH146<sup>LexA</sup>, lexAop-mRFP-nls/UAS DTA; VT033008-Gal4, UAS-CD8GFP/+ |
| | GH146<sup>LexA</sup>, lexAop-mRFP-nls/P(caryP)attp40; VT033008-Gal4, UAS-CD8GFP/+ |
| *Figure 5D–F* | UAS-DTA/+; VT033008-Gal4/TM6B |
| | UAS-DTA/+; P(caryP)attp2/TM6B |
| *Figure 5G–H* | GH146<sup>LexA</sup>, lexAop-mRFP-nls/UAS DTA; VT033006-Gal4, UAS-CD8GFP/+ |
| | GH146<sup>LexA</sup>, lexAop-mRFP-nls/P(caryP) attp40; VT033006-Gal4, UAS-CD8GFP/+ |
| *Figure 5I–L* | MB247-DsRed/UAS DTA; VT033006-Gal4, UAS-CD8GFP/+ |
| | MB247-DsRed/+; VT033006-Gal4, UAS-CD8GFP/+ |

| Figure 5—figure supplement 1A | GH146<sup>LexA</sup>, lexAop-mRFP-nls/UAS DTA; VT033008-Gal4, UAS-CD8GFP/+ |
|---|---|
| | GH146<sup>LexA</sup>, lexAop-mRFP-nls/CyO; VT033008-Gal4, UAS-CD8GFP/+ |
| Figure 5—figure supplement 1B | GH146<sup>LexA</sup>, lexAop-mRFP-nls/UAS DTA; VT033006-Gal4, UAS-CD8GFP/+ |
| | GH146<sup>LexA</sup>, lexAop-mRFP-nls/CyO; VT033006-Gal4, UAS-CD8GFP/+ |
| Figure 5—figure supplement 2 | MB247-DsRed/UAS DTA; VT033006-Gal4, UAS-CD8GFP/+ |
| | MB247-DsRed/+; VT033006-Gal4, UAS-CD8GFP/+ |
| Figure 6 | GH146<sup>LexA</sup>, LexAop-SPA-T2A-SPA/UAS DTA; VT033006-Gal4/+ |
| | GH146<sup>LexA</sup>, LexAop-SPA-T2A-SPA/P(caryP)attp40; VT033006-Gal4/+ |
| Figure 7, Figure 7—figure supplement 1, Figure 7-videos 1, 2 | UAS-GCaMP6s/UAS-GCaMP6s; UAS-GCaMP6s/UAS-GCaMP6s; OK107<sup>Gal4</sup>/ OK107<sup>Gal4</sup> or + |

| Figure 5—figure supplement 1A | $GH146^{LexA}$, lexAop-mRFP-nls/UAS DTA; VT033008-Gal4, UAS-CD8GFP/+ |
|---|---|
| | $GH146^{LexA}$, lexAop-mRFP-nls/CyO; VT033008-Gal4, UAS-CD8GFP/+ |
| Figure 5—figure supplement 1B | $GH146^{LexA}$, lexAop-mRFP-nls/UAS DTA; VT033006-Gal4, UAS-CD8GFP/+ |
| | $GH146^{LexA}$, lexAop-mRFP-nls/CyO; VT033006-Gal4, UAS-CD8GFP/+ |
| Figure 5—figure supplement 2 | MB247-DsRed/UAS DTA; VT033006-Gal4, UAS-CD8GFP/+ |
| | MB247-DsRed/+; VT033006-Gal4, UAS-CD8GFP/+ |
| Figure 6 | $GH146^{LexA}$, LexAop-SPA-T2A-SPA/UAS DTA; VT033006-Gal4/+ |
| | $GH146^{LexA}$, LexAop-SPA-T2A-SPA/P(caryP)attp40; VT033006-Gal4/+ |
| Figure 7, Figure 7—figure supplement 1, Figure 7-videos 1, 2 | UAS-GCaMP6s/UAS-GCaMP6s; UAS-GCaMP6s/UAS-GCaMP6s; $OK107^{Gal4}$/ $OK107^{Gal4}$ or + |

## HU ablation

Protocol was adapted from *Sweeney et al. (2012)*. Briefly, we set up large populations of flies in cages two days prior to ablation and placed a 35 or 60 mm grape juice agar plate (Lab Express, Ann Arbor, MI) in the cage with a dollop of goopy yeast. One day prior to the ablation, we replaced the grape juice/yeast plate with a new grape juice/yeast plate. On the morning of the ablation, we removed the plate from the cage and discarded the yeast puck and any hatched larvae on the agar. We then monitored the plate for up to four hours, until many larvae had hatched. Larvae were washed off the plate using a sucrose solution, and eggs were discarded. Larvae were then strained in coffee filters, and submerged in hydroxyurea (Sigma, H8627) in a yeast:AHL mixture, or sham mixture without HU. Ablation conditions were as follows:

*Figure 1*: 10 mg/mL or 5 mg/mL, 4 hr
*Figure 2A-E* 10 mg/mL, 1 hr
*Figure 2F-K*, *Figure 2—figure supplements 2*, *3*: 10 mg/mL, 1 hr; 3 or 5 mg/mL, 4 hr
*Figure 7*, *Figure 7—figure supplement 1*: 10 mg/mL, 1 hr

Larvae were then strained through coffee filters again, rinsed, and placed in a vial or bottle of R food until eclosion. We opened a new container of hydroxyurea each month as it degrades in contact with moisture and we found its potency gradually declined. We typically analyzed 5–10 animals (10–20 hemispheres) of each condition per batch, except in in vivo imaging experiments, where we analyzed 1–5 animals per condition per batch.

Applying HU for four hours produced a somewhat U-shaped distribution, with many samples unaffected and many with all four KC neuroblasts lost. Shifting from 3, to 5, to 10 mg over this long time scale shifted how many brains were completely ablated versus unaffected, but did little to increase the proportion of brains with intermediate phenotypes. Applying the stronger concentration, 10 mg, for a shorter time produced more sporadic effects, with more brains showing 1, 2, or 3 KC neuroblasts remaining. lNB/BAlc was more affected by HU concentration than time of application, with stronger effects seen applying 10 mg/mL for one hour than applying 3 or 5 mg/mL for four hours. Overall, combining these different protocols allowed us to observe a broad range of mushroom body states.

## Immunostainings

Brains were dissected for up to 20 min in external saline (108 mM NaCl, 5 mM KCl, 2 mM CaCl2, 8.2 mM MgCl2, 4 mM NaHCO3, 1 mM NaH2PO4, 5 mM trehalose, 10 mM sucrose, 5 mM HEPES pH7.5, osmolarity adjusted to 265 mOsm), before being transferred to 1% paraformaldehyde in PBS, on ice. All steps were performed in cell strainer baskets (caps of FACS tubes) in 24 well plates, with the brains in the baskets lifted from well to well to change solutions. Brains were fixed overnight at 4C in 1% PFA in PBS. On day 2, brains were washed 3 × 10' in PBS supplemented with 0.1% triton-x-100 on a shaker at room temperature, blocked 1 hr in PBS, 0.1% triton, 4% Normal Goat Serum, and then incubated for at least two overnights in primary antibody solution, diluted in PBS,

0.1% triton, 4% Normal Goat Serum. Primary antibody was washed 3 × 10' in PBS supplemented with 0.1% triton-x-100 on a shaker at room temperature, then brains were incubated in secondary antibodies for at least two overnights, diluted in PBS, 0.1% triton, 4% Normal Goat Serum. When used, Alexa 568-conjugated phalloidin (1:80) and/or DAPI (one microgram/mL) were included in secondary antibody mixes. Primary antibodies used were mouse anti-ChAT 9E10 (DSHB, 1:200), Rabbit anti-dsRed (Clontech, 1:500-1:1000), Sheep anti-GFP (Bio-Rad, 4745–1051, 1:250-1:1000), Chicken anti-GFP (1:5000, gift from Dawen Cai). Secondary antibodies were Alexa 488, 568, and 647 conjugates (1:500, Invitrogen).

Brains were mounted in 1x PBS, 90% glycerol supplemented with propyl gallate in binder reinforcement stickers sandwiched between two coverslips. Samples were stored at 4C in the dark prior to imaging. The coverslip sandwiches were taped to slides, allowing us to perform confocal imaging on one side of the brain and then flip over the sandwich to allow a clear view of the other side of the brain. This allowed us to score features on the anterior and posterior sides of each sample. Scanning confocal stacks were collected along the anterior-posterior axis on a Zeiss 880 or Leica SP8 with one micrometer spacing in Z and ~200 nm axial resolution.

## Statistical considerations

Determination of sample size: Brains were prepared for imaging in batches of 5–10. In initial batches, we assessed the variability of the manipulation, for example if we were trying to change Kenyon cell number, we looked at how variable the size of the Kenyon cell population was following the manipulation. We used this variability to determine how many batches to analyze so as to obtain enough informative samples. To avoid introducing statistical bias, we did not analyze the effect of the manipulation on the other cell type until after completing all batches; for example if the manipulation was intended to alter Kenyon cell numbers, we did not assess projection neuron bouton phenotypes until completing all samples. Genotypes or conditions being compared with one another were always prepared for staining together and imaged interspersed with one another to equalize batch effects, and we used at least two batches for each type of experiment.

Criteria for exclusion, treatment of outliers: In *Figures 1–6*, we only excluded from analysis samples with overt physical damage to the cells or structures being measured. In figures and analyses we treated outliers the same way as other data points. For *Figure 7*, full criteria for inclusion and exclusion of each sample are shown under *Analysis of KC somatic odor responses*, below.

Recent discussions among experts have suggested that biological studies over-rely on statistical tests, report p values when statistical differences are self-evident, and emphasize statistical significance rather than effect size (*Goedhart, 2018a*; *Goedhart, 2018b*; *Ho et al., 2019*). These choices can cloud findings instead of clarifying them. In order to communicate our findings in the simplest and most complete way, we have displayed each data point for each sample to allow readers to assess effect size and significance directly.

## Image analysis

In general, we analyzed mixed-sex populations, where sex ratios were carefully balanced across experimental conditions. Because MZ19 labels the sexually dimorphic DA1 PNs, in MZ19 experiments we analyzed the two sexes independently in pilot experiments. As we observed no correlation with sex (not shown), in subsequent MZ19 experiments we used mixed sexes.

Researchers performing quantification could not generally be blinded to experimental condition due to the overt changes in neuron numbers and brain structures induced by our manipulations. However, analysis was performed blind to the goals of the experiment when possible, and quantitation of features on the anterior and posterior sides of the brain were recorded independent of one another and merged after all quantifications were completed. Moreover, many of our analyses make use of variation within an experimental condition or genotype, providing an additional bulwark against observational bias.

To measure neuropil structures such as the mushroom body calyx, lateral horn, or antennal lobe, we used markers such as ChAT and phalloidin to visualize the structure, identified its largest extent in Z (i.e. along the A-P axis), outlined it in FIJI (as in white outlines in Figure 1B) and then calculated the cross-sectional area using the 'Measure' command.

To analyze calyx volumes we used ImageJ. Briefly, we drew Regions of Interest (ROIs) around the MB247 (*Figure 1*) or OK107 (*Figure 3*) calyx signal for each slice. Other imaging channels displaying projection neurons and nc82 staining were used as a reference. The drawn ROIs for each slice were saved using the ROI manager plugin and the area of each ROI in a stack was measured using the Measure tool. The measured areas were added together and then multiplied by the Z-spacing between slices (one micron) to get the volume of the combined ROI through the stack. The 3D viewer plugin was used to make movies of the kenyon cell and calyx volumes. To produce 3D Movies of the calyx volumes without somata, areas outside the selected ROIs were cleared and then the 3D viewer plugin was used to display only the 3D ROI volume that was measured for the volume analysis. To facilitate comparison in 3D movies, stacks in a set were all given the same depth in Z by adding blank slices at the end of the stack when necessary.

To count aggregate boutons, we used ChAT signal with reference to phalloidin (which stains actin-rich structures, including KC dendrites), except in *Figure 1* and *Figure 3A*, where we counted GH146-positive boutons. We counted as separate structures ChAT signals that were compact and appeared discontinuous with one another and that were 2+ micrometers in diameter (*Figure 1—figure supplement 2*). When phalloidin signal or KC fluorescence was available, two boutons would be counted separately if ChAT signals were separated by phalloidin signal or KC signal. In initial analyses, we found that boutons in slice 0 often appeared in slices −1 and +1 as well, but never in slices −2 or +2. In order to avoid counting the same boutons more than once, we therefore began counting at the most superficial slice in the stack where boutons were visible, and counted every other slice, i.e. every second micron. To count boutons of small groups of PNs, we drove fluorescent reporters under control of 42D01, 71D09, or MZ19 and counted coherent and compact fluorescent signals, with reference to ChAT signals and to phalloidin, when available. As GH146 labels ~2/3 of PNs, our counts of 450–650 total GH146+ boutons in *Figures 1*, *3*, which would correspond to 675–975 total boutons per calyx, correspond with estimates established by EM (578 boutons) and light microscopy (768 total boutons, or five boutons per PN) (*Leiss et al., 2009*; *Turner et al., 2008*; *Zheng et al., 2018*).

To count cell populations, we used genetically-encoded fluorescence as indicated. We counted labeled somata in every third slice in the stack (every third micron along the A-P axis), with reference to DAPI to distinguish individual cells from one another. As we did for boutons, in analyzing somata we initially determined that somata in slice 0 could also be seen in slices −2, −1, +1, and +2 but not in slice −3 or +3. To avoid double-counting, we therefore counted every third micron.

To count KC claws, we scanned image stacks for cup-shaped terminal structures of the appropriate size (1–2 μm in diameter). We sometimes labeled small groups of KCs. We scored independently any single KC or small group of KCs that could be visually separated from one another (like the 3 cells shown in *Figure 6C*). For *Figure 2* and *Figure 2—figure supplement 3*, we did not score any cells which we could not separate from one another because we did not want to mix counts for KCs of different types. For *Figure 3I*, where we did not attempt to separate KCs of different types, and *Figure 6F*, where our dye labeling strategy was specific to αβ KCs, we present an average claw/KC data point for small groups of 2–4 cells whose dendritic structures were interwoven. Finally, in *Figure 3I*, we found that in calyces with expanded KC populations, KC axons sometimes could not enter the pedunculus and instead wandered around at the base of the calyx. As CNS neurons in *D. mel* are unipolar, these wandering axons could be difficult to distinguish from dendritic claws. To determine whether these wandering axons should be counted as claws, we stained a subset of photoactivated brains with anti-ChAT (not shown), to determine which labeled KC neurites were in the bouton region of the calyx, and should thus be scored for claws, and which were 'lost' axons. Though photoactivated GFP signals were only poorly preserved following staining, relating ChAT-stained images to images of the same brain by two photon allowed us to learn to identify the bouton region of the calyx in subsequent experiments due to its 'swiss cheese' appearance, where the presence of PN boutons produces holes in KC fluorescence.

To determine KC and PN neuroblast state–PN lNB/BAlc: Whenever possible, we included GH146 or 60A10 driving a fluorescent report in our animals and scored the presence or absence of a group of somata lateral to the antennal lobe on the anterior side of the brain for each analyzed hemisphere. In *Figure 2F–H*, we examined ChAT+ somata ventrolateral of the antennal lobe to determine the presence or absence of lNB/BAlc progeny. KC neuroblasts: In *Figure 2A–C*, we used 58F02 to fluorescently label late-born KCs and counted clumps of labeled somata surrounding the

calyx as well as groups of labeled neurites leaving the calyx and entering the pedunculus. These estimates usually matched; in the few cases where they did not, we used the number of axon clumps, as somata are closer to the surface of the brain and more susceptible to mechanical disruption during dissection. As seen in *Figure 2B*, 58F02 labels additional neurons; these were easy to discriminate from KCs as they did not enter the calyx or pedunculus.

For *Figure 2J,K* and *Figure 2—figure supplement 3C,D*, we combined data from two different types of experiments. In some experiments, we expressed PA-GFP in all KCs using MB247. We traced the axons of labeled KCs into the lobes to assign KC type (*Figure 2—figure supplement 3*). In a second group of experiments, we used 58F02-Gal4 to express PA-GFP exclusively in αβ core KCs (*Figure 2I*), or 89B01 Gal4 to express PA-GFP only in γ KCs and thus knew KC type a priori. 89B01 also labels γd KCs, which can be distinguished from olfactory γ KCs as they do not innervate the main calyx. We do not include γd KCs in our analysis.

GFP photoactivation and Texas red dextran dye labeling were performed as previously described (*Clowney et al., 2015*; *Ruta et al., 2010*), using a Bruker Investigator microscope and Spectra-Physics MaiTai laser with DeepSee module. Dye filling electrodes were pulled using a Brown/Flaming puller (Sutter Instruments) and were guided to the mushroom body α lobes using visible light illumination and/or two photon autofluorescence.

## Identification of driver lines for relevant cell populations

To identify Gal4 and/or LexA lines labeling PN populations, we screened the Janelia FlyLight collection online database and then crossed lines with potential to fluorescent reporters. While 42D01 (~10 PNs) and 60A10 (60 PNs) were the most useful PN lines for us here, we also identified other lines, 33C10 and 37H08, that each label small groups of PNs and may be of use to others. To identify Gal4 lines labeling individual KC types, we screened expression of 'regular' FlyLight Gal4 constructs made using the same regulatory fragments incorporated in KC-type-specific split Gal4s (*Aso et al., 2014*). We found 58F02 useful for labeling α/β core KCs, 89B01 for γ KCs (labels both γmain and γd), and 41C07 for α'/β' KCs. 89B01 and 58F02 LexA did not recapitulate Gal4 expression patterns driven by the same fragments.

## Flow Cytometry

Brains were dissected in Schneider's medium (Sigma S0146) supplemented with 1% BSA and placed on ice. After all dissections were completed, collagenase (Sigma C0130) was added to a final concentration of 2 mg/mL and samples were incubated at 37C for 20 min. Samples were dissociated by trituration and spun down at 300g, 4C, for 5 min. Collagenase solution was removed and replaced with PBS+0.1% BSA supplemented with 1:1000 DyeCycle Violet. Cells were incubated with dye for 30 min on ice and passed through a cell strainer cap, before being subjected to flow cytometry on an Attune Flow Cytometer. Forward scatter and side scatter measurements were used to gate single cells, and DyeCycle Violet signal was used to gate cell bodies from membrane debris.

## In vivo functional imaging

We prepared 2–7 day old adult flies subjected to HU or sham treatment just after larval hatching for in vivo two photon calcium imaging on a Bruker Investigator essentially as described previously (*Ruta et al., 2010*), affixing the fly to packaging tape (Duck EZ Start) with human hair and UV glue (Loctite 3106). The fly was tilted to allow optical access to KC somata, at the dorsal posterior surface of the brain. In some experiments, we waxed the proboscis in an extended position to reduce motion. Imaging was performed in external saline. For odor delivery, half of a 1200 mL/min airstream, or 600 mL/min, was directed toward the antennae through a Pasteur pipette mounted on a micromanipulator. At a trigger, 25% of the total air stream was re-directed from a 10 mL glass vial containing mineral oil to a vial containing odorants diluted 1:10 in mineral oil, or a second vial of mineral oil (mechanosensory control). Final odor dilution was therefore 1:40. Filter paper 'wicks' were inserted into each vial to allow odor to saturate the vial headspace. Odors were delivered for two seconds, with 30–60 s in between stimulations. We used a simplified olfactometer capable of delivering five different odorants in which overall airflow was metered by analogue flowmeters (Brooks Instruments) and valve switching controlled by an Arduino. Odor delivery was initially

optimized using a mini-PID placed in the half of the air stream not directed at the fly (Aurora Biosciences). Images were collected at 1.5–5 Hertz, and we imaged a single plane for each sample.

## Analysis of KC somatic odor responses

We collected data from a total of 14 sham-treated animals (28 hemispheres imaged) and 24 HU-treated animals (46 hemispheres imaged). The two hemispheres were imaged separately. Occasionally, only one hemisphere was imaged due to the preparation and a few hemispheres were excluded from analysis because of poor image quality. Mineral oil was delivered first, and then each of four odors was delivered twice, in sequence. Following mineral oil, odor order varied. Because experiments in locust found that the first presentation of certain odors can cause distinct KC responses to all subsequent presentations of that same odor, we analyzed the second presentation of each odor, according to convention (*Murthy et al., 2008*; *Stopfer and Laurent, 1999*). We also observed that in some animals, enduring increases in fluorescence followed the first presentation of isobutyl acetate.

Following functional imaging, we either collected a Z-stack of the mushroom body on the two-photon, or dissected out the brain and subjected it to immunostaining and confocal imaging. We used these images to categorize the extent of KC reduction. All samples are accounted for here:

|      | Total hem | Poor image | Damaged/ not responsive | Unknown | Full ablation | Can't tell if ablated | Doesn't look ablated | Too much motion | Analyzed |
|------|-----------|------------|-------------------------|---------|---------------|----------------------|----------------------|-----------------|----------|
| Sham | 28        | 2          | 8                       | 1       | N/A           | N/A                  | N/A                  | 10              | 7: four fem, three male |
| HU   | 46        |            | 4                       | 0       | 13            | 5                    | 10                   | 7               | 7: four male, three fem |

We excluded 11 hemispheres from the control dataset due to damage or poor image quality. We also excluded 34 hemispheres from the ablation dataset because of any of the following: the samples were fully ablated, unaffected by ablation, ablation status was ambiguous, or the mushroom body calyx appeared damaged. The remaining samples were motion-corrected using Suite2p (*Pachitariu et al., 2017*). Using FIJI, we blinded ourselves to the remaining datasets and excluded datasets if there was too much motion to be able to follow the same cell over time. After this gate, there were seven sham and seven reduced-KC hemispheres for the analysis. ROIs were chosen in each of the fourteen samples and we measured $\Delta f/f$ (i.e. $(f-f_0)/f_0$), comparing stimulus frames with pre-stimulus frames. ROIs were manually chosen and we were careful about ensuring that the cell remained within its ROI for all frames. For two of the ablated samples, we had to choose different ROIs for two of the odor deliveries due to motion. These samples were only used to calculate the proportion of cells that were responsive to each odor, and not in calculating the number of odors each cell responded to. The baseline was determined for each odor delivery by taking the average of the frames within the three seconds before the trigger. The stimulus frame was determined by taking the peak response between 0–4.5 s after stimulus onset, to account for the valve opening and the two second odor delivery. To define 'responsive' cells, we chose to use the same cutoff, $\Delta f/f > 0.2$ (20% increase in fluorescence over baseline) across samples. This was in order to avoid obscuring overall differences in responsiveness across ablated and sham-treated calyces. For visualization, we used the Pretty Heatmap package in R to cluster cells with similar odor responses and display them.

## Acknowledgements

We thank the Bloomington Drosophila Stock Center, Yoshi Aso, and Tzumin Lee for providing fly strains; Vanessa Ruta for intellectual and material support during the early phases of this project; Isaac Levine, Shiloh Boosamra, Noella Holmer, Joseph Carter, and James Harrison for technical assistance; Shyama Nandakumar for help with flow cytometry experiments; Christina May, Swathi Yadlapalli, Ari Zolin, Yoshi Aso, Ashok Litwin-Kumar, Laura Buttitta, and members of the Cheng-Yu Lee lab for experimental advice and discussion; and Cathy Collins, Monica Dus, Rich Hume, Vanessa Ruta, and members of the EJC lab for comments on the manuscript. This work was supported by the

Simons Fellowship of the Helen Hay Whitney Foundation to EJC. EJC is the Rita Allen Foundation Milton Cassel Scholar and is supported by the Alfred P Sloan Foundation Award, and by a Neuroscience Scholar Award and start-up funds from the University of Michigan.

## Additional information

### Funding

| Funder | Grant reference number | Author |
| --- | --- | --- |
| Rita Allen Foundation | Milton Cassel Scholar | E Josie Clowney |
| Alfred P. Sloan Foundation | Research Fellow in Neuroscience | E Josie Clowney |
| University of Michigan | Startup Funds | E Josie Clowney |
| University of Michigan | Neuroscience Scholar Award | E Josie Clowney |
| Helen Hay Whitney Foundation | Simons Fellow | E Josie Clowney |

The funders had no role in study design, data collection and interpretation, or the decision to submit the work for publication.

### Author contributions

Najia A Elkahlah, Jackson A Rogow, Formal analysis, Investigation, Visualization, Methodology, Writing - review and editing; Maria Ahmed, Formal analysis, Writing - review and editing; E Josephine Clowney, Conceptualization, Resources, Formal analysis, Supervision, Funding acquisition, Investigation, Visualization, Methodology, Writing - original draft, Project administration, Writing - review and editing

### Author ORCIDs

Najia A Elkahlah (iD) https://orcid.org/0000-0002-9657-1777
Jackson A Rogow (iD) https://orcid.org/0000-0001-5060-6770
Maria Ahmed (iD) https://orcid.org/0000-0003-0054-3315
E Josephine Clowney (iD) https://orcid.org/0000-0002-9150-9464

### Decision letter and Author response

Decision letter https://doi.org/10.7554/eLife.52278.sa1
Author response https://doi.org/10.7554/eLife.52278.sa2

## Additional files

### Supplementary files

• Transparent reporting form

### Data availability

Analyzed data points generated during the study are included in the figures. Source data for Figure 7 is presented in a source data file.

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
