## [Decision Letter]

**Acceptance summary:**

Your paper uses the olfactory circuit of *Drosophila* to test developmental plasticity and robustness in a stochastic sparse wiring circuit between olfactory projection neurons and Kenyon cells in the mushroom body. This sparse wiring encodes olfactory perception. Your simple, well designed and elegant experiments revealed that there is significant presynaptic developmental plasticity explaining the robustness of sparse wiring. By manipulating cell divisions of either Kenyon cells or projection neurons to increase or decrease their population size and measuring "claws" (their connections), you realized that the number of post-synaptic sites of Kenyon cells is invariant while the number of presynaptic sites of projection neurons changes accordingly.

**Decision letter after peer review:**

Thank you for submitting your article "Presynaptic developmental plasticity allows robust sparse wiring of the *Drosophila* mushroom body" for consideration by *eLife*. Your article has been reviewed by Catherine Dulac as the Senior Editor, a Reviewing Editor, and three reviewers. The following individuals involved in review of your submission have agreed to reveal their identity: Bassem A Hassan (Reviewer #3).

The reviewers have discussed the reviews with one another and the Reviewing Editor has drafted this decision to help you prepare a revised submission.

The three reviewers agree about the importance of the work and find the problem and the way it is addressed to be highly significant: The principle of developmental plasticity in presynaptic neurons in this circuit is fascinating and the way you manipulate it is very good!

However, as you will see from the comments, two of the three reviewers have some serious issues with the counting of buttons that are presented in Figure 7. They agree that this is not a trivial task (although others have done so very well, see Leiss, 2009) but they are not convinced by the experimental data that you show and they found it difficult to verify the accuracy of the counting, which is critical to the paper.

Therefore, we would like you to provide more information on how you did the counting and to show precise images with markers indicating what is counted as a bouton, and what is not. In particular, you need to elaborate about how you avoided counting errors across optical sections. Furthermore, the n number is too low to allow statistics and might need to be improved.

Reviewer #1:

The authors examine the rules for wiring the input connections in the mushroom body, what sets the degree of synaptic convergence of PN inputs onto KCs. They address this by manipulating cell divisions of either KC or PN progenitors to increase or decrease population size. Based on measurements of KC claws and calyx cross-section, the number of KC post-synaptic sites appears invariant across conditions, while the number of PN presynapses changes.

There are some important technical points that are not adequately described and may not be adequate to justify the conclusions. Figure 7 is based on a seriously insufficient data set. Without resolving these issues, I am not confident in their results:

1) They quantify the size of the calyx as the maximum cross-sectional area. They should quantify calyx volume instead. There could be large change in calyx volume without changes in max cross-sectional area.

The Materials and methods section description of this should also be clarified. It says: '…identified its [calyx's] largest extent in z, outline it in FIJI…'. Do they mean they found the confocal plane with the widest cross section and calculated the area there? They should use brain-based coordinates (i.e. dorso-ventral/A-P) rather than Z since I don't know how they're scanning the brain. Also, it sounds like they simply measure a diameter and assume the calyx is a circle. If so, that is too crude a measure. They should make a more precise measurement of the 3D boundary of the calyx and calculate volume.

2) It's unclear how they actually counted PN boutons/cell bodies/claws. Again, the Materials and methods section description should be more accurate. It says ' counted every other slice' and 'counted every third slice' How do we know whether there is or isn't overlap between slices? We'd need to know the axial resolution, as well as the distance between slices but these are not reported. As written, it sounds like they made an arbitrary decision to skip counting some slices. It's impossible to tell if they could be double counting or missing many counts.

In Figure 1E the number of PN boutons a little less than 500, which is low compared to published results, Leiss et al., 2009 counted between 780 and 1600 depending on markers used, and Turner et al., 2008 estimated 1165 based on PN labeling. Also, the numbers in Figure 1E don't match their statement of '1000/calyx' on in the Materials and methods section. Can the authors please clarify?

Figure 2B shows examples of how they count the number of KC neuroblasts using the 58F02 driver. However, there are sometimes clusters of green cells that are apparently not counted e.g. left panel shows one big cluster and two smaller ones, second from left shows one cluster at 6'oclock and one at 9o'clock but this is assessed to be only one KC Nb. This needs more explanation. Is the driver not completely KC specific? How did they know that a cluster is made of up KCs and not other cells, especially with the 9 o'clock cluster.

3) The authors must take more measurements of KC odor responses in ablated animals, n=3 is not sufficient for any conclusions. In fact, there are no statistical tests in Figure 7 to support any conclusion.

What does the df/f value shown in e.g. Figure 7C represent? Is it peak df/f during the odor, or area under the curve or what?

It would be far better to show some example plots of df/f versus time (as a simple line plot rather than heatmap). This is a better way for the reader to evaluate the data, in particular to assess movement artefacts. Presenting essentially a single point, as in Figure 7C, can mask experimental issues.

The authors also use an arbitrary threshold to define a response: 20% df/f. This is not particularly well-justified and not standard in the field. In any case, there's no need for a response threshold to analyze these data. A better analysis would be to measure df/f values across the entire set of KCs and plot those distributions for ablated and sham animals. Viewing those two distributions is far more informative than a% response, which apparently ranges from 5-40% by the current criterion, so it's not a very robust measurement anyway. (In fact, in Figure 7D I see a number around 0.5, very high!)

The authors should be more forthcoming in their description of these results. Subsection “Developmental plasticity preserves sparse odor coding despite perturbations to cell populations” says 'variation is similar to previous reports' but the Materials and methods section says that more responses were observed in this study than previous, both referring to Honegger, 2011. The authors should be straightforward about the differences between the observations in both Main Text and Materials and methods section. The analysis suggested above should be adequate support for their claims (with enough n), since they are comparing sham and ablated.

Related to overall quality of these experiments: in Figure 7C are there instances of negative df/f values in the responses? It is hard to tell with the colormap the authors use, but I see it ranges down to -0.5. If so, that should be commented on, since inhibition is rarely seen with GCaMP. The concern is that movement artefacts are likely to give negative df/f values, so that should be ruled out.

Additionally, the high level of responsiveness the authors observe relative to previous work could also be due to movement artefacts giving artificially large df/f values. The authors should analyze some non-odor period of their calcium signals to see if similar response frequencies are observed. And do the same for inhibitory responses (if those are in fact negative values).

Reviewer #2:

The manuscript "Presynaptic developmental plasticity allows robust sparse wiring of the *Drosophila* mushroom body" by Elkahlah et al., uses various alteration in the ratios of neurons that contribute to the calyx of the *Drosophila* mushroom body to address how the convergence of combinatorial inputs is established during development. The idea put forth by the authors is that the convergence ratio is set by post-synaptic Kenyon cells-such that Kenyon cells produce relatively invariant numbers of claws, whereas the number of pre-synaptic specializations on Projection Neurons can vary bi-directionally.

I very much like the question and approach described in this manuscript. I especially like that the authors took their analysis all the way to circuit function. However, much of the primary anatomical data is not always convincing. And there are experimental oversights that are either problematic or it is not fully explained why they are not problematic.

Figure 1. The authors variably reduce the number of Kenyon Cell neuroblasts and Projection Neuron neuroblasts using chemical ablation approaches. They show this leads to reduction of Kenyon Cell number and reductions in projection neuron bouton numbers.

Can the authors show projection neuron boutons at higher resolution? I have no idea how they counted the boutons from these images. Much of their argument hinges on their ability to count these structures, so showing this in a convincing way in Figure 1 is important.

Figure 2. In the same manipulation as Figure 1, the authors score for additional phenotypes-In panel F, how do these images allow one to count boutons? The images are too low-resolution to tell, could they be displayed bigger, or without the overlay? I can't tell how the drawings are the right correspond to the images that are shown. Similarly, but more dramatically, for the claws in Figure 2F, especially the bottom panel, I cannot understand how the raw image data gives the authors the schematic that they show at the right. I want to believe their quantified data, but seeing these types of raw data make me question!

Figure 3. The authors knock down Mud to expand the Kenyon Cell neuroblast pool. Figure 3C

-Are the neurons properly specified? Are there molecular markers that could be used confirm neuronal identity? (Similar comments are true for PN data in Figure 4).

-Why is it of no concern to the authors that OK107 drives mud-RNAi in mature neurons?

-Subsection “Olfactory projection neurons increase bouton repertoire as Kenyon cell number increases” of the text says that the MZ19+ boutons are doubled and cites Figure 3C-D. There is no expansion show in Figure 3C-D. What is Figure 3E What are AA and B above the panel on the right? (Similar annotations show elsewhere were confusing).

In Figure 3F, I can't see the claws. These images are not high-enough resolution.

Figure 4. The authors knock down Mud to expand the Projection Neuron neuroblast pool. I cannot understand how the authors counted projection neuron neuroblast number. In Figure 4C, it looks like there is an increased density of boutons?

Figure 5. On the y-axis of 5C and 5G what does "+" mean? Does it mean "both" or "expressing"?

In Figure 5F, what is the residual green?

Do the authors not think it is important to know when they kill cells with DTA?

Reviewer #3:

Elkahlah and colleagues use the PN-MB olfactory circuit in flies to test the limits of developmental plasticity and robustness in a stochastic sparse wiring circuit, where the sparsity of wiring is thought to be critical for encoding.

I truly enjoyed reading this manuscript. The authors perform a number of simple, well designed and elegant experiments to address the question. They find that there is significant presynaptic developmental plasticity explaining the robustness of sparse wiring. I particularly love the approach of testing an idea through clever manipulation of neuronal numbers showing that one does not necessarily need a "molecular mechanism" to understand a basic principle of how the brain is wired.

I have no major comments and think that the paper can be published essentially as it is.

---

## [Author Response]

However, as you will see from the comments, two of the three reviewers have some serious issues with the counting of buttons that are presented in Figure 7. They agree that this is not a trivial task (although others have done so very well, see Leiss, 2009) but they are not convinced by the experimental data that you show and they found it difficult to verify the accuracy of the counting, which is critical to the paper.Therefore, we would like you to provide more information on how you did the counting and to show precise images with markers indicating what is counted as a bouton, and what is not. In particular, you need to elaborate about how you avoided counting errors across optical sections. Furthermore, the n number is too low to allow statistics and might need to be improved.

In reading through the reviews, we think the summary about methods of bouton counting is related to Figure 1, Figure 2, Figure 3, Figure 4, Figure 5, Figure 6, as Figure 7 has no bouton counts. We think the comment related to n is related to in vivo imaging, in Figure 7.

As described below, we have provided more information and supplemental figures showing how we count the boutons, as well as summarizing the existing literature. We have also increased the sample size in Figure 7 from 3 animals per condition to 7 animals per condition.

Reviewer #1:The authors examine the rules for wiring the input connections in the mushroom body, what sets the degree of synaptic convergence of PN inputs onto KCs. They address this by manipulating cell divisions of either KC or PN progenitors to increase or decrease population size. Based on measurements of KC claws and calyx cross-section, the number of KC post-synaptic sites appears invariant across conditions, while the number of PN presynapses changes.There are some important technical points that are not adequately described and may not be adequate to justify the conclusions. Figure 7 is based on a seriously insufficient data set. Without resolving these issues, I am not confident in their results:1) They quantify the size of the calyx as the maximum cross-sectional area. They should quantify calyx volume instead. There could be large change in calyx volume without changes in max cross-sectional area.The Materials and methods section description of this should also be clarified. It says: '…identified its [calyx's] largest extent in z, outline it in FIJI…'. Do they mean they found the confocal plane with the widest cross section and calculated the area there? They should use brain-based coordinates (i.e. dorso-ventral/A-P) rather than Z since I don't know how they're scanning the brain. Also, it sounds like they simply measure a diameter and assume the calyx is a circle. If so, that is too crude a measure. They should make a more precise measurement of the 3D boundary of the calyx and calculate volume.

We apologize for the vagueness in our methods. At the outset of this project, we tested a number of different methods for measuring calyx size in order to find an analysis approach that was both accurate and feasible given the large number of samples to be quantified. We found that for KC ablation experiments (Figures1, Figure 2), the maximum cross-sectional area of the calyx served as an accurate proxy for the number of Kenyon cells. As we now clarify in the methods, this area is determined by drawing a precise outline around the calyx at its maximum extent along the A-P axis and measuring the area inside this shape. Ovals such as shown in Figure 1 were not used for quantification. To give a better sense of calyx dimensions following ablation, we now show volume contours for the images in Figure 1B (Figure 1—video 1), and measure the volumes of these five samples (Figure 1—figure supplement 1). As calyx max cross-sectional area is correlated with both KC number and with calyx volume (Figure 1—figure supplement 1), we are confident that it serves as a relevant metric.

We also quantified the number of total boutons in our images. Except in Figure 5 and Figure 6, where we ablate olfactory projection neurons, we did not observe alterations in bouton size or shape, or empty areas of the calyx. Bouton number therefore provides a functionally meaningful measure of alterations across the whole volume of the calyx. In Figure 5 and Figure 6, we indeed describe that calyx area is no longer a good proxy for calyx contents.

Please note: In Figure 1—video 1, we have not been able to force the rendered mushroom bodies to be displayed at the same scale, though a standard-sized bounding box (red) is displayed on each sample. This is due to the time frame of the revision and the software available to us. So as not to delay the submission of the revision, we are including the video as is. We will include it or exclude it from the published paper as advised.

2) It's unclear how they actually counted PN boutons/cell bodies/claws. Again, the Materials and methods section description should be more accurate. It says ' counted every other slice' and 'counted every third slice' How do we know whether there is or isn't overlap between slices? We'd need to know the axial resolution, as well as the distance between slices but these are not reported. As written, it sounds like they made an arbitrary decision to skip counting some slices. It's impossible to tell if they could be double counting or missing many counts.In Figure 1E the number of PN boutons a little less than 500, which is low compared to published results, Leiss et al., 2009 counted between 780 and 1600 depending on markers used, and Turner et al., 2008 estimated 1165 based on PN labeling. Also, the numbers in Figure 1E don't match their statement of '1000/calyx' on in the Materials and methods section. Can the authors please clarify?

Thank you to the reviewers for calling attention to discrepancies in bouton counts and in the number of boutons reported in the literature. We will summarize all comments related to bouton counts here.

a) How many boutons are there actually in the calyx?

We have reviewed the prior literature about bouton number and think that 600-800 is likely to be the correct number, with 1600 an overestimate. First, recent EM data reports 578 boutons/calyx, less than the estimates from Leiss et al., 2009 or Turner et al., 2008. Second, the estimate in Turner of 1115 boutons is made by extrapolating from the number of boutons on individual PNs to an estimated population of 216 PNs. More recent estimates suggest there are 150 or fewer PNs (Frechter et al., 2019; Zheng et al., 2018), which would reduce the estimate from Turner to ~800 total boutons. Third, in Leiss, the average number of boutons identified by manual counting is 768. When the authors overexpress a LimK-GFP fusion protein, they are able to count boutons “semiautomatically” but obtain more than double the number, ~1600. LimK is an actin binding protein, and Kenyon cell dendrites are actin-rich structures, suggesting that bouton number and morphology may not be wild type in these animals. Indeed, the authors suggest that the high number of boutons in this condition may be due to changes in bouton morphology caused by overexpression of LimK. Together with the three other quantifications (Turner, Zheng, and our analysis here), as well as the uncertainty about the effects of LimK overexpression, it is likely that 1600 is the outlier, and the true bouton number is closer to 600-800. Together, these results suggest that while the rule of thumb in the field is that there are ~1000 boutons/calyx, the great majority of calyces likely have less than this number. Indeed, we did not observe 1000 boutons/calyx in our own data; our statement on line 859 was incorrect.

a) What did we count?

In Figure 1 and Figure 3, we counted not total boutons (stained with ChAT), but GH146-positive boutons. GH146 labels ~2/3 of total PNs. We stated this in the legend of Figure 1 but were in error in describing Figure 3 as counting ChAT-positive boutons. We have now fixed Figure 3 and made more clear that in Figure 1 we count GH146-positive boutons. Given that we count 450-700 GH146+ boutons, the total number in the calyx would be ~1.5-fold higher, or 675-1050. This is a reasonable number of total boutons given the prior literature.

a) How did we count the boutons?

Reviewer 1 suggests that our bouton counts were arbitrary. This is far from the case! We collected confocal stacks from anterior to posterior with ~200nm axial resolution (depending on the wavelength) and 1μm slices in Z. We have moved this information from the Materials and methods section closer to our description of bouton counting to allow more clarity. We then counted all boutons in every second image (a census every two microns), as described in the original manuscript. When we counted *somata*, we counted every third micron.

In initiating our analysis, we scanned through our confocal stacks to define how far boutons extended across images. We found that boutons visible in slice 0 were often visible in slices -1 and +1, but never in slices -2 and +2. Specifically, to avoid overcounting, we therefore counted all boutons every second micron. While this may undersample the boutons, undersampling is not statistically problematic a priori. Moreover, we definitely did not double-count boutons, which would be statistically problematic. Most important of all for our purposes is that the control and experimental datasets are sampled equivalently, which our methods ensured. We now include a supplemental methods image showing what we count as boutons from a particular confocal slice (Figure 1—figure supplement 2).

We tried, both previously and in response to the reviewer comments, to count as is described in Leiss, “…constantly moving through the Z-stack.” We found that making this assessment of when to count and when not to count a bouton allowed too much room for experimenter interpretation and bias and was therefore less consistent than counting every second micron. However, for 71D09 experiments, where we count boutons of just a single PN, we were able to confirm that scanning up and down gave the same results as counting every second micron.

Importantly, we were not able to measure boutons in any automated way, though we tried various software packages, and the authors in Leiss were only able to use automated methods to count boutons when bouton morphology became more orderly, through LimK overexpression in Kenyon cells.

Figure 2B shows examples of how they count the number of KC neuroblasts using the 58F02 driver. However, there are sometimes clusters of green cells that are apparently not counted e.g. left panel shows one big cluster and two smaller ones, second from left shows one cluster at 6'oclock and one at 9o'clock but this is assessed to be only one KC Nb. This needs more explanation. Is the driver not completely KC specific? How did they know that a cluster is made of up KCs and not other cells, especially with the 9 o'clock cluster.

Yes, the reviewer is correct to point out that the driver does not label Kenyon cells exclusively, and that the green clumps not highlighted are not Kenyon cells. We defined what signals corresponded to Kenyon cells by following them into the mushroom body pedunculus. We have now clarified this in the legend of Figure 2B.

3) The authors must take more measurements of KC odor responses in ablated animals, n=3 is not sufficient for any conclusions. In fact, there are no statistical tests in Figure 7 to support any conclusion.

In Figure 7, we have now more than doubled our dataset, so that we present 7 sham and 7 reduced-KC samples. We include statistical tests and additional analyses in Figure 7 and Figure 7—figure supplement 1. We note that to our knowledge, this is the first analysis that attempts to define effects on neural function of developmental perturbations of this circuit, and that the number of sham-treated animals we present is consistent with the number of animals used in previous studies to draw conclusions about Kenyon cell function in unmanipulated animals.

In this context, our analyses have not been exhaustive. While many aspects of odor representation appear preserved in reduced-KC calyces, we also observe statistically significant differences in responses to some odors when cells are pooled across animals for analysis. It is difficult to decide whether each cell should “count” as a biological replicate, or each animal, and of course our statistical power to detect changes is very different depending on that choice. There may also be additional quantitative differences between conditions that we cannot capture even with our increased sample size, due to the high variability across control animals (which is expected given that Kenyon cells are randomly innervated, and corresponds with prior literature). We hope we have been appropriately circumspect in our language about what we can and cannot learn given our data.

What does the df/f value shown in e.g. Figure 7C represent? Is it peak df/f during the odor, or area under the curve or what?

This is the peak df/f; we have now clarified this throughout.

It would be far better to show some example plots of df/f versus time (as a simple line plot rather than heatmap). This is a better way for the reader to evaluate the data, in particular to assess movement artefacts. Presenting essentially a single point, as in Figure 7C, can mask experimental issues.

We now show such traces over time in Figure 7—figure supplement 1, and provide supplemental videos (Figure 7—video1, Figure 7—video 2) of imaging datasets.

The authors also use an arbitrary threshold to define a response: 20% df/f. This is not particularly well-justified and not standard in the field. In any case, there's no need for a response threshold to analyze these data. A better analysis would be to measure df/f values across the entire set of KCs and plot those distributions for ablated and sham animals. Viewing those two distributions is far more informative than a% response, which apparently ranges from 5-40% by the current criterion, so it's not a very robust measurement anyway. (In fact, in Figure 7D I see a number around 0.5, very high!)

Thank you for this suggestion. In Figure 7D of the revised manuscript, we now show such an analysis. We also now present df/f values for all our analyzed cells and samples (Figure 7—source data 1) such that others can perform their own analyses.

We were also surprised that so many Kenyon cells responded to isobutyl acetate and ethyl acetate, given rules of thumb in the field that ~10% of KCs respond to a given odor. We are using a more sensitive GCaMP than prior imaging studies (GCaMP6s versus GCaMP3 and GCaMP1.3), so we may detect more responses than in previous work. There is also variation across studies in the odor concentrations used (from 1:10 to 1:1000; we used 1:40), and variation in composition of the bath saline, which will also affect neuronal responsiveness. It is therefore reasonable that the proportion of cells responding to odor would be different across studies.

We performed a systematic review of studies that report the responses of individual Kenyon cells to a variety of odors. Responsiveness reported in electrophysiological datasets varies depending on the spiking criteria used to define “responsive,” as discussed in Murthy, Fiete and Laurent, 2008, from a maximum of 18% of KCs called as “responding” to a given odor in Turner, 2008, versus up to 75% in Murthy. Moreover, proportion of Kenyon cells responding to odors increases along with the sensitivity of the GCaMP variant used, from a maximum of 2% of cells responding to a particular odor with GCaMP1 (Wang, 2004), to a maximum of 20% of cells with GCaMP3 (Honegger 2013, Campbell 2013), and up to 80% responding with GCaMP6s (our results). Our results with GCaMP6s correspond well with the spiking criteria defined in Murthy, and are also consistent with the expectation that GCaMP1 and GCaMP3 will underrepresent neuronal activity compared to GCaMP6s.

StudyMethodSalineOdor ConcentrationKCs respondingWang 2004GCaMP1.3 in OK107“adult fly saline”1:1000.1% to 2%Murthy 2008Whole cell patch clampMurthy1:1025%-75%Murthy 2008Whole cell patch clampMurthy1:10017%-37%Murthy 2008Whole cell patch clampMurthy1:10000%-50%Turner 2008Whole cell patch clampWilson1:10000%-18%Campbell 2013GCaMP3 in OK107Wilson1:1003%-17%Honegger 2013GCaMP3 in OK107Wilson1:1001%-20%Elkahlah 2019GCaMP6s in OK107External saline1:400%-80%

Wilson saline (in mM): NaCl 103, KCl 3, TES 5, trehalose 10, glucose 10, sucrose 7, NaHCO3 26, NaH2PO4 1, CaCl2 1.5, MgCl2 4 (adjusted to 280 mOsm). The saline was bubbled with 95% O2/5% CO2 and continuously perfused over the preparation (2ml/min).

Murthy saline (in mM): NaCl 103, KCl 3, TES 5, NaHCO_3_ 26, NaH_2_PO_4_ 1, CaCl_2_ 1.5, MgCl2 4, trehalose 10, glucose 10, sucrose 9 (pH = 7.25, 275 mOsm). The saline was bubbled with 95% O_2_ and 5% CO_2_ and continuously perfused over the preparation.

External saline (Clowney): 108 mM NaCl, 5 mM KCl, 2 mM CaCl2, 8.2 mM MgCl2, 4 mM NaHCO3, 1 mM NaH2PO4, 5 mM trehalose, 10 mM sucrose, 5 mM HEPES pH7.5, osmolarity adjusted to 265 mOsm.

“adult fly saline” (Wang): 115 mm NaCl, 5 mm KCl, 6 mm CaCl_2_·2H_2_O, 1 mm MgCl2·6H_2_O, 4 mm NaHCO_3_, 1 mM NaH_2_PO_4_·1H_2_O, 5 mm trehalose, 75 mm sucrose, and 5 mm *N-*Tris (hydroxymethyl) methyl-2-aminoethanesulfonic acid, pH 7.1, 356 mOsm]

The authors should be more forthcoming in their description of these results. Subsection “Developmental plasticity preserves sparse odor coding despite perturbations to cell populations” says 'variation is similar to previous reports' but the Materials and methods section says that more responses were observed in this study than previous, both referring to Honegger, 2011. The authors should be straightforward about the differences between the observations in both Main Text and Materials and methods section. The analysis suggested above should be adequate support for their claims (with enough n), since they are comparing sham and ablated.

We specifically made a distinction here between absolute proportion of cells responding, which indeed is higher in our work compared to Honegger, versus the high levels of variation in cell-wise responsiveness seen across different animals, which is consistent between our analysis and Honegger.

Related to overall quality of these experiments: in Figure 7C are there instances of negative df/f values in the responses? It is hard to tell with the colormap the authors use, but I see it ranges down to -0.5. If so, that should be commented on, since inhibition is rarely seen with GCaMP. The concern is that movement artefacts are likely to give negative df/f values, so that should be ruled out.Additionally, the high level of responsiveness the authors observe relative to previous work could also be due to movement artefacts giving artificially large df/f values. The authors should analyze some non-odor period of their calcium signals to see if similar response frequencies are observed. And do the same for inhibitory responses (if those are in fact negative values).

Indeed, movement artifacts can be very difficult to eliminate in in vivo imaging in flies, especially in imaging closely packed and tiny somata (a “bowl of grapes” problem). We have made changes in our experimental preparation, imaging, and analysis to reduce these artifacts. First, in the new samples included here, we have reduced motion in the preparation by waxing the proboscis in an extended conformation prior to imaging. Second, we started to collect data at a higher frame rate (5 Hz), and to collect images continuously during the rest periods between odors trials. These two changes improve Suite2p motion correction. Third, for both the new samples presented here and the samples presented in the original submission, we have now included only ROIs in analysis for which the cell does not leave the ROI during the odor trial. As can be seen in the new Figure 7—figure supplement 1, our cell-wise traces over time look stable and responses are time-locked to odor stimulus. While a few motion artifacts are still present (e.g. see “MCH sham” panel in that figure), these are clearly distinguishable from odor-driven changes.

Reviewer #2:The manuscript "Presynaptic developmental plasticity allows robust sparse wiring of the Drosophila mushroom body" by Elkahlah et al., uses various alteration in the ratios of neurons that contribute to the calyx of the *Drosophila* mushroom body to address how the convergence of combinatorial inputs is established during development. The idea put forth by the authors is that the convergence ratio is set by post-synaptic Kenyon cells-such that Kenyon cells produce relatively invariant numbers of claws, whereas the number of pre-synaptic specializations on Projection Neurons can vary bi-directionally.I very much like the question and approach described in this manuscript. I especially like that the authors took their analysis all the way to circuit function. However, much of the primary anatomical data is not always convincing. And there are experimental oversights that are either problematic or it is not fully explained why they are not problematic.Figure 1. The authors variably reduce the number of Kenyon Cell neuroblasts and Projection Neuron neuroblasts using chemical ablation approaches. They show this leads to reduction of Kenyon Cell number and reductions in projection neuron bouton numbers.Can the authors show projection neuron boutons at higher resolution? I have no idea how they counted the boutons from these images. Much of their argument hinges on their ability to count these structures, so showing this in a convincing way in Figure 1 is important.

Thank you for pointing this out, we now show higher resolution images in Figure 1B, as well as a supplemental figure, Figure 1—figure supplement 2, showing what counts as a bouton during quantification.

Figure 2. In the same manipulation as Figure 1, the authors score for additional phenotypes-In panel F, how do these images allow one to count boutons? The images are too low-resolution to tell, could they be displayed bigger, or without the overlay? I can't tell how the drawings are the right correspond to the images that are shown. Similarly, but more dramatically, for the claws in Figure 2F, especially the bottom panel, I cannot understand how the raw image data gives the authors the schematic that they show at the right. I want to believe their quantified data, but seeing these types of raw data make me question!

Yes, we now show higher-resolution images in Figure 2F and point to what counts as a bouton and have removed the schematics

Figure 3. The authors knock down Mud to expand the Kenyon Cell neuroblast pool. Figure 3C-Are the neurons properly specified? Are there molecular markers that could be used confirm neuronal identity? (Similar comments are true for PN data in Figure 4).-Why is it of no concern to the authors that OK107 drives mud-RNAi in mature neurons?-Subsection “Olfactory projection neurons increase bouton repertoire as Kenyon cell number increases” of the text says that the MZ19+ boutons are doubled and cites Figure 3C-D. There is no expansion show in Figure 3C-D. What is Figure 3E What are AA and B above the panel on the right? (Similar annotations show elsewhere were confusing).In Figure 3F, I can't see the claws. These images are not high-enough resolution.

We do think the neurons are properly specified, as they continue to express the KC marker OK107, and PNs continue to express the marker GH146. (While these are enhancer traps, not antibodies, they are both integrated into loci, eyeless and oaz, that are important for patterning these cell types.) The fact that these neurons still contact each other is itself a reflection of their proper specification. Moreover, these neurons innervate other appropriate targets-supernumerary PNs innervate the antennal lobe and lateral horn, in addition to the calyx, and KCs innervate the mushroom body axonal lobes except in the most extreme expansions.

We have RNAseq data for wild type PNs and KCs at 45h APF and in adults. We show here that there is *no* endogenous expression of *mud* in either cell type at either time point, which makes us comfortable using an RNAi that is not restricted to neuroblasts. We think it is overkill to add these 9 RNAseq datasets to the publication, but are open to doing so if desired. To generate this data, we labeled KCs with MB247, and PNs with GH146. We dissociated neurons as already described in our FACs-analysis methods, and FAC-sorted them, retaining PNs, KCs, and unlabeled cells (“Double Negative” or “DN”). We prepared RNA using SmartSeq V2, and subjected libraries to deep sequencing. We include below *Oaz* and *ey*, markers of PNs and KCs, respectively. Adult data is an average of two replicates; 45h APF data is from a single library for each cell type. Numbers here are FPKM generated by Cufflinks/Cuffdiff.

DN45hPN45hKC45hDNadult (Avg)PNadult (Avg)KCadult (Avg)mud0.02234680.039413700.005883490.01086540*nsyb*337.596230.206155.922740.2042628.74787.3Oaz0.16764966.01460.1284230.25497171.6220ey5.421951.21215227.7887.577510.143108275.23l(2)gl1.045422.1538400.3767960.5273450.301086aPKC37.24537.6717643.212881.0263185.60193.7232pros19.957212.40618.5323748.9148152.8460.9881

We cannot completely rule out a role for *mud* in neuronal identity/specification, but we are more comfortable using *mud* than other neuroblast factors which continue to be expressed in differentiating neurons, e.g. aPKC, l(2)gl, and pros shown above. We continue to search for additional methods to expand these populations.

We also note that because expansion of these populations is sporadic, many of the mud-RNAi brains have indistinguishable neuron numbers from wild type and olfactory circuits that appear indistinguishable from wild type despite *mud* knockdown. This also suggests that *mud* knockdown does not affect specification.

Subsection “Olfactory projection neurons increase bouton repertoire as Kenyon cell number increases” —The KC expansions we obtained from the genotype with the MZ19 PNs labeled were more subtle than in other experiments; we are not sure why this is the case. However, we obtained 4 brains with calyces larger than any of the controls (>2000µM). One of these is shown in the right panel of Figure 3C (Figure 3D in the revision). As shown in Figure 3F, calyces with maximum cross-sectional area greater than 2000µM have an average of ~100 MZ19+ boutons, versus ~55 in controls. The letters above these graphs are the statistically same and different groups from ANOVA, which we have clarified in the legend. (i.e. groups with the same letter are statistically indistinct, while groups with different letters are statistically distinct).

Figure 4. The authors knock down Mud to expand the Projection Neuron neuroblast pool. I cannot understand how the authors counted projection neuron neuroblast number. In Figure 4C, it looks like there is an increased density of boutons?

We did not count PN NB number, but rather number of PNs. Because of the described role for *mud* in neuroblast asymmetric division, we assume change in PN number results from change in PN NB number.

The increased density in red in Figure 4C is due to thickened PN axon tract passing over the calyx in this maximum intensity Z-projection, which we have now clarified in the text. As shown in the ChAT staining in blue, the density of boutons is similar to control.

Figure 5. On the y-axis of 5C and 5G what does "+" mean? Does it mean "both" or "expressing"?In Figure 5F, what is the residual green?Do the authors not think it is important to know when they kill cells with DTA?

We include a new figure, Figure 5—figure supplement 1, where we image reporter expression driven by VT033006 and VT033008, and effects of using these lines to drive DTA, in third-instar larvae. As can be seen in those panels, these drivers are “on” at least this early, and the cells are lost when toxin is present at least this early. This is around 2.5 days before PNs and KCs synapse (around 60h APF), and ~5 days before adult eclosion.